# Recapitulating human cardio-pulmonary co-development using simultaneous multilineage differentiation of pluripotent stem cells

Wai Hoe Ng[1], Elizabeth K Johnston[1], Jun Jie Tan[2], Jacqueline M Bliley[1,3], Adam W Feinberg[1,3], Donna B Stolz[4], Ming Sun[4], Piyumi Wijesekara[1], Finn Hawkins[5], Darrell N Kotton[5], Xi Ren[1]*

[1]Department of Biomedical Engineering, Carnegie Mellon University, Pittsburgh, United States; [2]Advanced Medical and Dental Institute, Universiti Sains Malaysia, Penang, Malaysia; [3]Department of Materials Science and Engineering, Carnegie Mellon University, Pittsburgh, United States; [4]Center for Biologic Imaging, University of Pittsburgh, Pittsburgh, United States; [5]Center for Regenerative Medicine of Boston University and Boston Medical Center, Boston, United States

**Abstract** The extensive crosstalk between the developing heart and lung is critical to their proper morphogenesis and maturation. However, there remains a lack of models that investigate the critical cardio-pulmonary mutual interaction during human embryogenesis. Here, we reported a novel stepwise strategy for directing the simultaneous induction of both mesoderm-derived cardiac and endoderm-derived lung epithelial lineages within a single differentiation of human-induced pluripotent stem cells (hiPSCs) via temporal specific tuning of WNT and nodal signaling in the absence of exogenous growth factors. Using 3D suspension culture, we established concentric cardio-pulmonary micro-Tissues (µTs), and expedited alveolar maturation in the presence of cardiac accompaniment. Upon withdrawal of WNT agonist, the cardiac and pulmonary components within each dual-lineage µT effectively segregated from each other with concurrent initiation of cardiac contraction. We expect that our multilineage differentiation model will offer an experimentally tractable system for investigating human cardio-pulmonary interaction and tissue boundary formation during embryogenesis.

*For correspondence:
xiren@cmu.edu

## Editor's evaluation

The study responded very well to the expert reviewers and offers new insights into mechanisms regulating differentiation of cardiac and pulmonary stem cells. It will stimulate further investigations into this important field.

## Introduction

Human embryogenesis is a highly orchestrated process that requires delicate coordination between organs that originate from different germ layers. As the two main organs within the chest cavity, the mesoderm-derived heart and endoderm-derived lung partake in have extensive mutual interaction that are essential for their proper morphogenesis (*Peng et al., 2013*; *Hoffmann et al., 2009*; *Arora et al., 2012*; *Steimle, 2018*). During mouse embryonic development, WNT derived from the second heart field induces specification of pulmonary endoderm, which in turn secretes SHH that

**eLife digest** Organs begin developing during the first few months of pregnancy, while the baby is still an embryo. These early stages of development are known as embryogenesis – a tightly organized process, during which the embryo forms different layers of stem cells. These cells can be activated to turn into a particular type of cell, such as a heart or a lung cell.

The heart and lungs develop from different layers within the embryo, which must communicate with each other for the organs to form correctly. For example, chemical signals can be released from and travel between layers of the embryo, activating processes inside cells located in the different areas.

In mouse models, chemical signals and cells travel between developing heart and lung, which helps both organs to form into the correct structure. But it is unclear how well the observations from mouse models translate to heart and lung development in humans.

To find out more, Ng et al. developed a human model of heart and lung co-development during embryogenesis using human pluripotent stem cells. The laboratory-grown stem cells were treated with chemical signals, causing them to form different layers that developed into early forms of heart and lung cells.

The cells were then transferred into a specific growing condition, where they arranged into three-dimensional structures termed microtissues. Ng et al. found that lung cells developed faster when grown in microtissues with accompanying developing heart cells compared to microtissues containing only developing lung cells. In addition, Ng et al. revealed that the co-developing heart and lung tissues automatically separate from each other during later stage, without the need for chemical signals.

This human cell-based model of early forms of co-developing heart and lung cells may help provide researchers with new strategies to probe the underlying mechanisms of human heart and lung interaction during embryogenesis.

signals back to the heart and regulates proper atrial septation (*Steimle, 2018*; *Zhou, 2017*; *Hoffmann et al., 2009*). This inter-lineage crosstalk is partly mediated by the multipotent mesodermal progenitors located between the developing heart and lung, which have the potential for lineage contribution to pulmonary endothelium, pulmonary smooth muscle and cardiomyocytes (*Peng et al., 2013*). However, the extent of translation of findings derived from rodent models to the understanding of developmental interplay between human cardio-pulmonary systems remains unclear. There is, therefore, a critical need for experimentally tractable systems for investigating human cardio-pulmonary co-development during organogenesis.

Much work has been done for directed differentiation of hiPSCs into either cardiomyocytes (*Lian et al., 2012*; *Mummery et al., 2012*; *Burridge et al., 2014*; *Lian et al., 2015*; *Lee et al., 2017*; *Kattman et al., 2011*), or pulmonary epithelium (*Chen et al., 2017*; *Huang et al., 2013*; *Jacob et al., 2017*; *Dye et al., 2015*; *Gotoh et al., 2014*; *Wong et al., 2012*), both of which often utilize stepwise differentiation strategies that recapitulate key developmental signaling events. To recapitulate cardiogenesis, hiPSCs were sequentially specified into mesoderm, cardiac mesoderm, and then NKX2.5$^+$ cardiac progenitors (*Lian et al., 2012*; *Mummery et al., 2012*; *Burridge et al., 2014*; *Lian et al., 2015*; *Lee et al., 2017*; *Kattman et al., 2011*; *Laflamme et al., 2007*). For pulmonary induction, hiPSCs went through stages corresponding to definitive endoderm and anterior foregut endoderm, and then became NKX2.1$^+$ lung epithelial progenitors (*Chen et al., 2017*; *Huang et al., 2013*; *Jacob et al., 2017*; *Dye et al., 2015*; *Gotoh et al., 2014*; *Wong et al., 2012*; *D'Amour et al., 2005*; *Longmire et al., 2012*). Despite the significant contributions of these models make to the mechanistic understanding of human heart and lung organogenesis, they generally focus on one organ parenchyma at a time. It remains challenging to model and investigate multi-organ co-development within a single differentiation of hiPSCs, especially when the organs of interest are derived from different germ layers, as is the case for the heart and lung.

Comparison of existing protocols for single-lineage cardiac and pulmonary differentiation from hiPSCs indicates shared regulators despite their distinct germ-layer origin. Firstly, both endodermal and mesodermal specification is facilitated by the inhibition of insulin and phosphoinositide 3-kinase

signaling (*Lian et al., 2012*; *Lian et al., 2013*; *Mou et al., 2012*), and can be induced by a similar set of paracrine factors, including WNT, BMP, and TGF-β (*Kattman et al., 2011*; *D'Amour et al., 2005*; *Loh, 2014*). It is the quantitative combination of these signaling that determines endoderm versus mesoderm bifurcation (*Kattman et al., 2011*; *Loh, 2014*; *Kim, 2015*). This is consistent with the shared primitive streak origin of both germ layers during gastrulation (*Levak-Svajger and Svajger, 1974*; *Lawson et al., 1991*; *Tam and Beddington, 1987*). Secondly, WNT inhibition not only facilitates the transition from definitive endoderm to anterior foregut endoderm (*Loh, 2014*; *Spence et al., 2011*), but it also promotes cardiac mesoderm emergence (*Lian et al., 2012*; *Willems et al., 2011*; *Wang et al., 2011*; *Ren et al., 2011*; *Tran, 2009*). Lastly, retinoic acid (RA) signaling is required for the induction and maturation of both cardiac and pulmonary progenitors (*Huang et al., 2013*; *Jacob et al., 2017*; *Mou et al., 2012*; *Chen, 2007*; *McCauley et al., 2017*). These common paracrine regulation of paralleled cardiac and pulmonary specification is consistent with their close spatial coordinates within the embryonic body planning, as demonstrated by shared HOX genes expression and functional requirement (*Lufkin et al., 1991*; *Makki and Capecchi, 2012*; *Chisaka and Capecchi, 1991*).

In this study, we described a stepwise, growth-factor-free protocol for simultaneous induction of cardiac and pulmonary progenitors from a single culture of hiPSCs. This is accomplished by initial co-induction of mesoderm and definitive endoderm mixture, followed by their concurrent specification into cardiac (NKX2.5$^+$) and lung (NKX2.1$^+$) progenitors, respectively, using the same sets of small molecule cocktails modulating WNT, nodal and TGF-β signaling in a temporal specific manner. Using 3D suspension culture with continuing WNT activation, we engineered pulmonary-centered, cardio-pulmonary micro-Tissues (µTs), and demonstrated the accompanying cardiac lineage as an essential cellular niche that promoted effective alveolar maturation. Finally, following the withdrawal of WNT agonist, each concentric cardio-pulmonary µT reorganized and ultimately segregated into cardiac-only and pulmonary-only µTs. This work therefore offers an effective hiPSC-based model for investigating cardio-pulmonary co-development and tissue segregation during human embryogenesis.

## Results

### Simultaneous induction of cardiac and pulmonary progenitors

Building on existing protocols on cardiac (*Lian et al., 2012*; *Mummery et al., 2012*; *Burridge et al., 2014*; *Lian et al., 2015*; *Lee et al., 2017*; *Kattman et al., 2011*; *Laflamme et al., 2007*), and lung (*Chen et al., 2017*; *Huang et al., 2013*; *Jacob et al., 2017*; *Dye et al., 2015*; *Gotoh et al., 2014*; *D'Amour et al., 2005*; *Green et al., 2011*) differentiation from hiPSCs, a stepwise differentiation strategy was developed to enable simultaneous specification of both lineages within a single culture of hiPSCs. Firstly, a balanced mesodermal and endodermal induction was achieved via fine-tuning of WNT activation in the absence of insulin, activin A, and BMP4 supplementation (Stage-1). Then, a combined inhibition of WNT and TGF-β signaling initiated the specification of the co-induced mesoderm and endoderm towards cardiac and pulmonary specification, respectively (Stage-2). Lastly, reactivation of WNT signaling in the presence of retinoic acid (RA) led to concurrent emergence of NKX2.5$^+$ cardiac and NKX2.1$^+$ lung progenitors (Stage-3), which was our main focus in this study.

WNT signaling is required for mesodermal and endodermal specification in dose-dependent manner during embryogenesis as well as hiPSC specification (*Lian et al., 2012*; *Kattman et al., 2011*; *Paige, 2010*; *Ng et al., 2008*). Using a hiPSC line, BU3 NKX2.1$^{GFP}$; SFTPC$^{tdTomato}$ (BU3-NGST), we examined the possibility of co-inducing mesodermal and definitive endodermal specification by exclusively modulating WNT signaling using a GSK3β inhibitor (CHIR99021, hereafter abbreviated as CHIR) and without the addition of exogenous growth factors (e.g. Activin A and BMP4). BU3-NGST were trypsinized and plated at 150,000 cells/cm$^2$ in mTESR1 medium for 24 hr. BU3-NGST were then treated with different concentrations of CHIR at 4, 7, and 10 µM for 48 hr in mTESR1 medium, followed by incubation in growth factor-free differentiation medium (based on RPMI1640 and B-27 minus insulin) (*Figure 1a*). Toward the end of germ layer induction (Stage-1), we detected co-existence of both definitive endodermal (SOX17$^+$) and mesodermal progenitors (MIXL1$^+$ and NCAM1$^+$), as well as the wide-spread expression of pan-mesendoderm marker (MIXL1$^+$) (*Figure 1b*). In addition, definitive endoderm co-expressed both FOXA2 and SOX17 (*Figure 1c*). This was further confirmed by flow cytometry analysis CD13 (mesodermal marker) and SOX17 (endodermal marker) (*Figure 1—figure supplement 1f-g*). This observation was further confirmed by gene expression analysis of FOXA2,

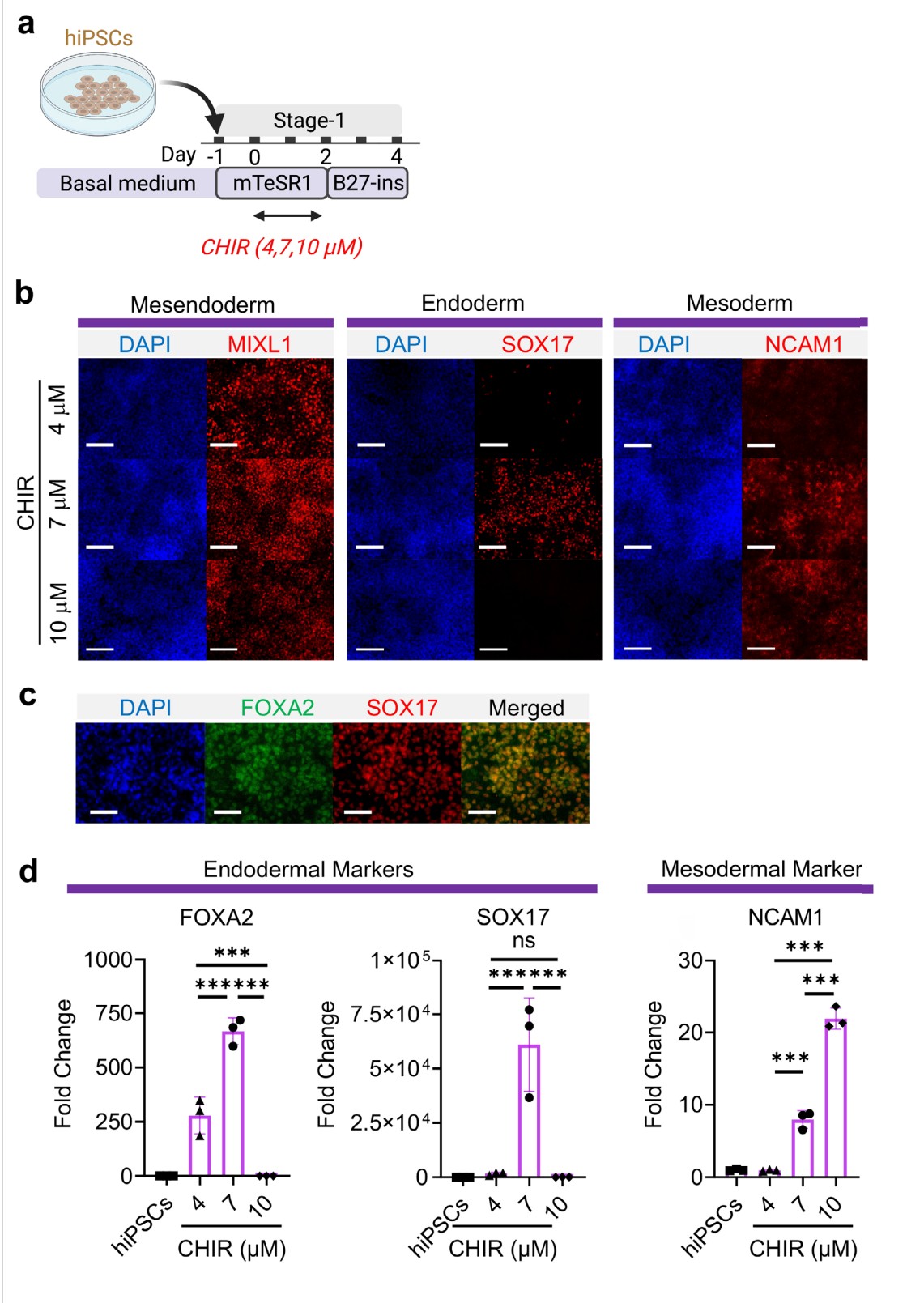

**Figure 1.** Mesoderm and endoderm co-induction from hiPSCs using CHIR. (**a**) Diagram showing the experimental design (**b**) Cells following Stage-1 differentiation expressed MIXL1 (Mesendodermal lineage), SOX17 (definitive endoderm), and NCAM1 (mesoderm). Scale bar = 125 μm. (**c**) Majority of SOX17 cells were also FOXA2$^+$. Scale bar = 62.5 μm. (**d**) Fold change of hiPSCs for FOXA2 (n = 3 each; 4 vs 7, p < 0.001; 7 vs 10, p < 0.001; 4 vs 10, p < 0.001), SOX17 (n = 3 each; 4 vs 7, p < 0.001; 7 vs 10, p < 0.001; 4 vs 10, p = 0.9978) and NCAM1 (n = 3 each; 4 vs 7, p < 0.001; 7 vs 10, p < 0.001; 4 vs

*Figure 1 continued on next page*

*Figure 1 continued*

10, p < 0.001). All data are mean ± SD. *p < 0.05; **p < 0.01; ***p < 0.001. 'n' refers to biological replicates. Diagram created using BioRender (http://biorender.com/).

The online version of this article includes the following source data and figure supplement(s) for figure 1:

**Source data 1.** Raw data for *Figure 1d*.

**Figure supplement 1.** Primitive streak induction from hiPSCs using CHIR.

SOX17, and NCAM1 (*Figure 1d*), which together suggests that 7 µM CHIR drives balanced endoderm and mesoderm induction from hiPSCs, while further elevation of CHIR dosage selectively favors mesodermal specification. This is in line with a study by *Martyn et al., 2019* demonstrating that WNT induces formation of primitive streak after 48 hr of treatment and that the newly formed primitive streak cells provide additional endogenous WNT, BMP, and Nodal signaling to further pattern the cells toward mesoderm and endoderm lineage (*Martyn et al., 2019*), We confirmed that upon 48 hr induction with 7 µM CHIR, the majority of cells were expressing primitive streak marker (T) and mesendoderm marker (MIXL1), but not pluripotent marker (OCT4) during early germ layer induction (*Figure 1—figure supplement 1*).

To specify the co-induced mesoderm and endoderm towards cardiac and pulmonary lineages, respectively, Day-4 cells were treated with TGF-β inhibitor (A8301) (*McCauley et al., 2017*; *Jacob et al., 2017*; *Hawkins et al., 2017*) and WNT inhibitor (IWP4) (*Lian et al., 2012*; *Huang et al., 2013*) for 4 days (Stage-2, Day-5 to Day-8), followed by treatment with a ventralization cocktail consisting of CHIR and RA (essential for lung progenitor specification) for 7 days to Day-15 (Stage 3) (*Figure 2a and b*; *McCauley et al., 2017*; *Jacob et al., 2017*; *Hawkins et al., 2017*). Consistent with CHIR-dependent germ layer induction (*Figure 1*), the efficiency of cardio-pulmonary specification was tightly regulated by CHIR dosage. We found that on Day-15, cells pre-exposed to CHIR (7 µM) during Stage-1 were able to give rise to robust co-induction of both cardiac (NKX2.5$^+$) and pulmonary (NKX2.1$^+$) progenitors (*Figure 2c, d and e*). In comparison, cells pre-treated with high-CHIR (10 µM) differentiated mainly into cardiac lineage; while low-CHIR (4 µM) failed to drive effective differentiation into either lineage (*Figure 2c, d and e*). FACS analysis confirmed that our protocol enabled effective induction of both NKX2.1$^+$ and NKX2.5$^+$ cells as compared to pulmonary-only or cardiac-only differentiation protocols (*Figure 2—figure supplement 1a, b, c*). Since NKX2.1 expression can also be found in neural and thyroid tissues, we performed further immunostaining analysis on Day-15 differentiated cells to inspect this possibility. Our results showed the NKX2.1$^+$ cells did not co-express TUJ1/PAX6 (Neural) or PAX8 (Thyroid), suggesting that the specified NKX2.1$^+$ population is of lung fate. Furthermore, no p63-expressing cells were identified, confirming the absence of airway epithelial cell population. At the same stage, NKX2.5$^+$ cells co-expressed cardiac Troponin T (cTnT), suggesting that these cells were being specified towards cardiac lineage.COUPTFII-positive staining was also observed in some of the NKX2.5$^+$ cells, suggesting atrial specification. However, most of the NKX2.5$^+$ cells have yet to specific into downstream cardiac subtypes such as ventricular (MLC2v), endocardium (NFATC), and epicardium (WT1) (*Figure 2—figure supplement 1d*).

The action of CHIR treatment on hiPSC differentiation depends not only on dosage but also the duration of exposure (*Kempf et al., 2016*; *Zhao et al., 2019*). We evaluated the efficiency of cardio-pulmonary induction following exposure to CHIR (7 µM) for different periods (24, 48, and 72 hr), and found that extended CHIR exposure for 48 or 72 hr was required to induce robust cardio-pulmonary programs (*Figure 2f*). Specifically, CHIR favored cardiac specification with increase in exposure time and plateaued at 48 hr of treatment (*Figure 2h*); while the induction of pulmonary program peaked at 48 hr of CHIR treatment (*Figure 2g*) and declined with further extension of the treatment. Based on these observations, for all subsequent experiments, we used 48 hr treatment of CHIR (7 µM) during Stage-1 of the co-differentiation program. Furthermore, we showed that maintaining hiPSCs in mTESR1 Plus during the initial CHIR treatment appeared to be critical for enabling effective cardio-pulmonary differentiation (*Figure 3—figure supplement 1*), as compared to using RPMI1640 supplemented with B-27 minus insulin as the basal medium during CHIR treatment.

Exogenous activation of nodal, TGF-β and BMP signaling during the very initial steps of hiPSC specification has been widely utilized for cardiac (*Kattman et al., 2011*; *Laflamme et al., 2007*; *Ng et al., 2008*) and pulmonary (*Huang et al., 2013*; *Gotoh et al., 2014*; *Mou et al., 2012*; *Jacob*

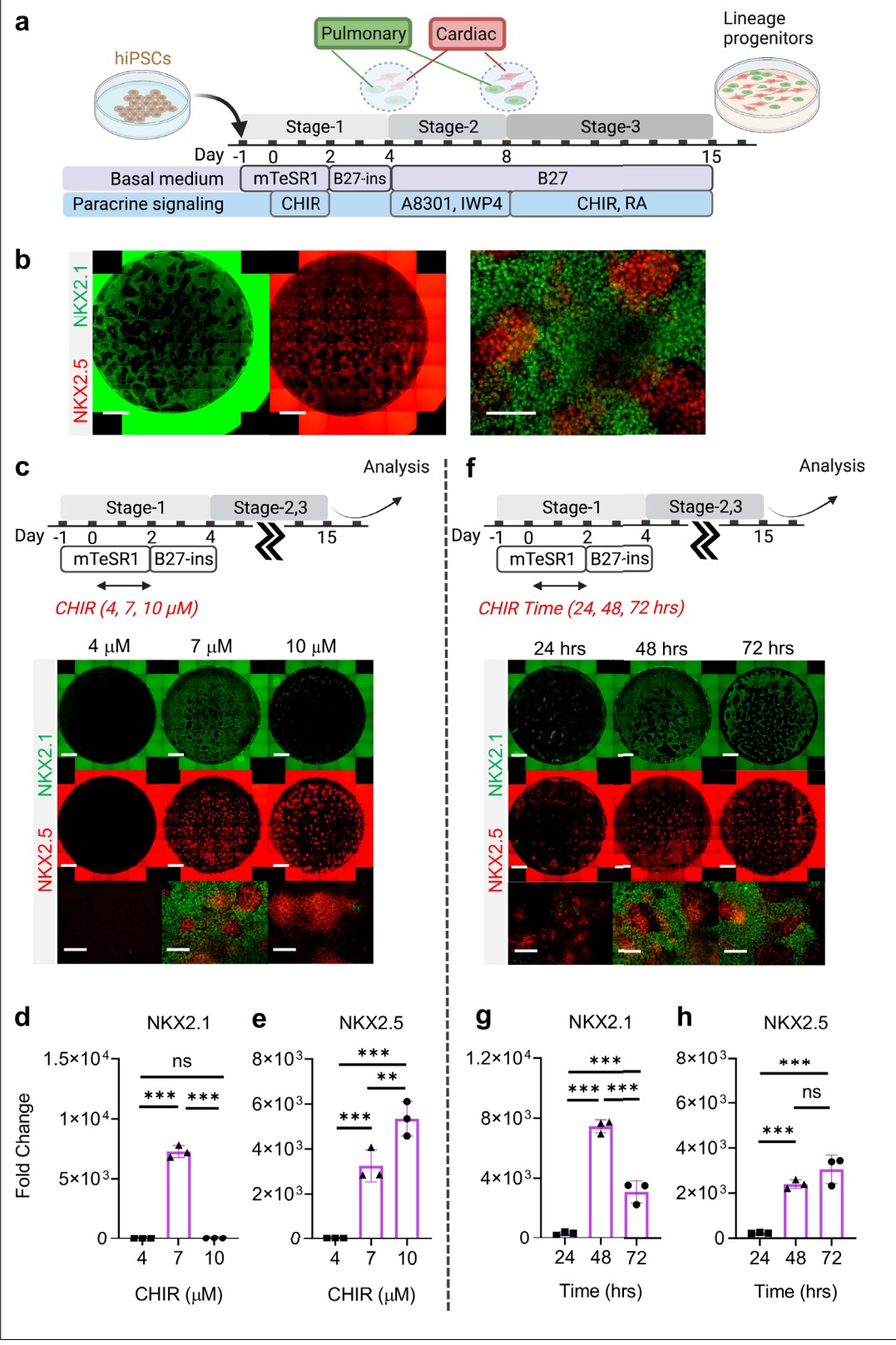

**Figure 2.** Stepwise cardio-pulmonary co-differentiation from hiPSCs using chemical defined, growth factor-free protocol. (**a**) Schematic diagram showing the overall differentiation strategy. (**b**) Immunofluorescence (IF) showing staining of lung (NKX2.1$^+$) and cardiac (NKX2.5$^+$). (**c**) IF (**d,e**) and quantitative PCR (qPCR) analysis of the induction of lung and cardiac progenitors on Day-15 of differentiation. (**c–e**) The effects of different CHIR concentrations

*Figure 2 continued on next page*

*Figure 2 continued*

during Stage-1 of differentiation. Fold change over hiPSCs (**d**) NKX2.1 (n = 3 each; 4 vs 7, p < 0.001; 7 vs 10, p < 0.001; 4 vs 10, p = 0.9993) and (**e**) NKX2.5 (n = 3 each; 4 vs 7, p< 0.001; 7 vs 10, p = 0.0053; 4 vs 10, p < 0.001). (**f–h**) The effects of different exposure time of CHIR (7 μM) treatment during the first 2 days of differentiation. qPCR analysis of (**g**) NKX2.1 (n = 3 each; 24 vs 48, p < 0.001; 48 vs 72, p < 0.001; 24 vs 72, p < 0.001) and (**h**) NKX2.5 (n = 3 each; 24 vs 48, p < 0.001; 48 vs 72, p = 0.1503; 24 vs 72, p < 0.001). Scale bar = 500 μm for whole well scan; Scale bar = 125 μm for 20 X images. All data are mean ± SD. *p < 0.05; **p < 0.01; ***p < 0.001. 'n' refers to biological replicates. Diagram created using BioRender (http://biorender.com/).

The online version of this article includes the following source data and figure supplement(s) for figure 2:

**Source data 1.** Raw data for *Figure 2d, e, g and h*.

**Figure supplement 1.** Characterization of Day-15 cardio-pulmonary progenitors.

*et al., 2017*; *Dye et al., 2015*) specification from hiPSCs. Here, we investigated how exogenous and endogenous nodal and BMP signaling regulates cardio-pulmonary induction during germ layer induction (Stage-1). Nodal signaling inhibition (using A8301, Day-2 to Day-4) immediately following CHIR treatment terminated both cardiac and pulmonary induction; while nodal activation through Activin A supplementation (Day-2 to Day-4) led to pulmonary-only differentiation (*Figure 3a, b and c*). This suggests the requirement of endogenous nodal signaling for cardio-pulmonary induction and that high-level nodal activation favors pulmonary instead of cardiac induction. In parallel, BMP inhibition (using DMH-1) during the same time period compromised cardiac induction and mildly reduced pulmonary specification; while exogenous BMP4 supplementation enhanced cardiac induction but inhibited pulmonary specification (*Figure 3d, e and f*). This indicates that endogenous BMP signaling is primarily required for cardiac induction and that exogenous augmentation of BMP signaling further favors the cardiac lineage at the expense of the pulmonary lineage.

## Shared signaling for cardio-pulmonary co-differentiation from germ-layer progenitors

In previous single-lineage hiPSC differentiation studies, TGF-β and WNT inhibition is known to promote pulmonary specification from definitive endoderm (*Huang et al., 2013*; *Gotoh et al., 2014*; *McCauley et al., 2017*; *Jacob et al., 2017*; *Hawkins et al., 2017*; *Dye et al., 2015*), as well as the induction of cardiac mesoderm (*Lian et al., 2012*). Here, we examined how combined inhibition of both TGF-β (using A8301) and WNT (using IWP4) during Day-4 to Day-8 (*Figure 3—figure supplement 2a*) regulates cardio-pulmonary specification from germ-layer progenitors established in Stage-1. We found that combined TGF-β and WNT inhibition enhanced both cardiac and pulmonary specification, with TGF-β inhibition having a more profound effect on the cardiac lineage (*Figure 3—figure supplement 2b,c,d*). Our finding suggests shared signaling requirement for lung and heart induction from their respective germ-layer progenitors, which is consistent with their close spatial coordinates within the embryonic body planning (*Lufkin et al., 1991*; *Makki and Capecchi, 2012*; *Chisaka and Capecchi, 1991*).

In both mouse and human pluripotent stem cell differentiation models, exogenous BMP4 has been shown to be crucial for ventralization of the foregut endoderm to give rise to NKX2.1[+] lung progenitors (*Huang et al., 2013*; *Jacob et al., 2017*; *Serra et al., 2017*). Here in our study, we observed effective cardio-pulmonary co-differentiation in the absence of exogenous BMP4 during ventralization (Stage 3) (*Figure 2b*). To address this discrepancy, we investigated how exogenous introduction of BMP4 during ventralization regulated the emergence of cardiac and pulmonary progenitors (*Figure 3—figure supplement 3a*). Intriguingly, there was no significant differences observed at protein and gene expression level of NKX2.1 and NKX2.5 comparing ventralization in the presence and absence of exogenous BMP4 (*Figure 3—figure supplement 3b,c,d*). Nonetheless, endogenous BMP4 was indeed required during this stage of differentiation, as inhibition of BMP4 using DMH1 significantly compromised the induction of both NKX2.1 and NKX2.5 (*Figure 3—figure supplement 3b,c,d*).

## 3D suspension culture platform for alveolar induction

To examine whether NKX2.1[+] lung progenitors derived from the cardio-pulmonary co-induction protocol (*Figure 2a*) possess the ability to mature into alveolar type 2 (AT2) epithelial cells, Day-15

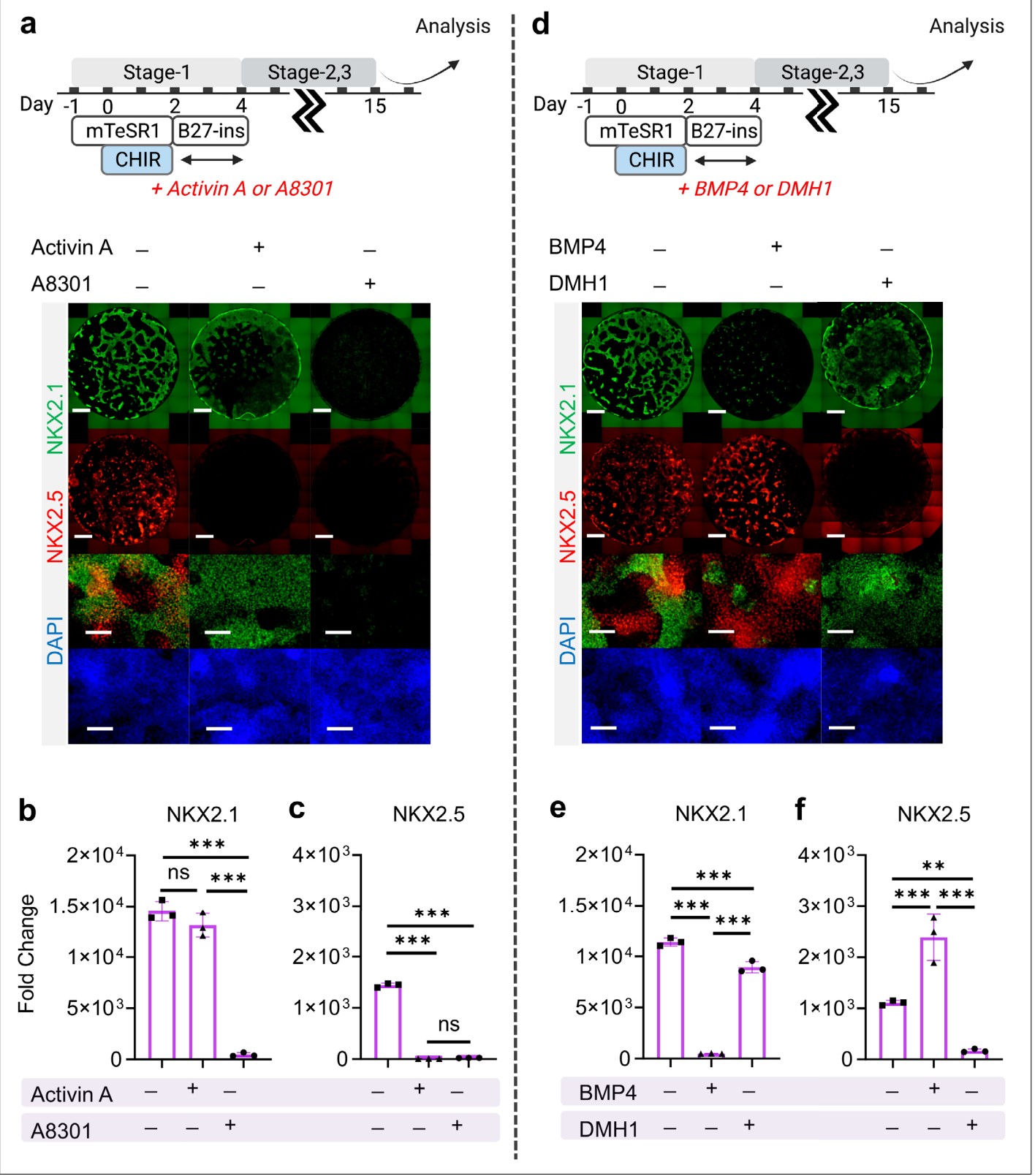

**Figure 3.** The effect of Nodal and BMP signaling during Stage-1 of co-differentiation on cardio-pulmonary induction. IF (**a,d**) and qPCR (**b,c,e,f**) analysis of the induction of lung (NKX2.1+) and cardiac (NKX2.5+) progenitors on Day-15 of differentiation (**a–c**) The effects of exogenous nodal activation (Activin A, 20 ng/mL) or its inhibition (A8301, 1 μM). Fold change over hiPSCs for (**b**) NKX2.1 (n = 3 each; Activin A− /A8301− vs. Activin A+ /A8301−, p = 0.1939; Activin A− /A8301− vs. Activin A− /A8301+, p < 0.001; Activin A+ /A8301− vs. Activin A− /A8301+, p < 0.001) and (**c**) NKX2.5 (n = 3 each; Activin A− /A8301− vs.

*Figure 3 continued on next page*

*Figure 3 continued*

Activin A⁺/A8301⁻, $p < 0.001$; Activin A⁻/A8301⁻ vs. Activin A⁻/A8301⁺, $p < 0.001$; Activin A⁺/A8301⁻ vs. Activin A⁻/A8301⁺, $p = 0.8649$). (**d-f**) The effects of exogenous BMP4 (20 ng/mL) or BMP inhibitor (DMH1, 2 μM). qPCR analysis of (**e**) NKX2.1 ($n = 3$ each; BMP4⁻/DMH1⁻ vs. BMP4⁺/DMH1⁻, $p < 0.001$; BMP4⁻/DMH1⁻ vs. BMP4⁻/DMH1⁺, $p < 0.001$; BMP4⁺/DMH1⁻ vs. BMP4⁻/DMH1⁺, $p < 0.001$) and (**f**) NKX2.5 ($n = 3$ each; BMP4⁻/DMH1⁻ vs. BMP4⁺/DMH1⁻, $p < 0.001$; BMP4⁻/DMH1⁻ vs. BMP4⁻/DMH1⁺, $p = 0.0044$; BMP4⁺/DMH1⁻ vs. BMP4⁻/DMH1⁺, $p < 0.001$). Scale bar = 500 μm for whole well scan; Scale bar = 125 μm for 20 X images. All data are mean ± SD. *$p < 0.05$; **$p < 0.01$; ***$p < 0.001$. 'n' refers to biological replicates. Diagram created using BioRender (http://biorender.com/).

The online version of this article includes the following source data and figure supplement(s) for figure 3:

**Source data 1.** Raw data for *Figure 3b, c, e and f*.

**Figure supplement 1.** Initial co-induction medium for CHIR-directed differentiation.

**Figure supplement 2.** Combination of TGF-β and WNT inhibition during Stage-2 of co-differentiation is required for cardio-pulmonary induction.

**Figure supplement 2—source data 1.** Raw data for *Figure 3—figure supplement 2c,d*.

**Figure supplement 3.** Roles of BMP4 during Stage-3 of co-differentiation.

**Figure supplement 3—source data 1.** Raw data for *Figure 3—figure supplement 3c,d*.

---

cells were trypsinized and re-plated into an ultra-low adhesion plate for 3D suspension culture (*Figure 4a*), and exposed to alveolar maturation medium containing **C**HIR, **K**GF, **D**examethasone, 8-bromoadenosine 3', 5'-cyclic monophosphate (**c**AMP activator), and **I**BMX (CKDCI) (*Jacob et al., 2017*; *Dye et al., 2015*; *de Carvalho et al., 2019*). Upon transition from 2D to 3D suspension culture in CKDCI medium, the co-induced cardio-pulmonary progenitors self-assembled into pulmonary-centered, concentric, dual-lineage μTs during the overnight culture (*Figure 4b*). Following 3 days of 3D suspension culture in CKDCI medium, effective AT2 maturation was observed in the cardio-pulmonary μTs as indicated by robust SFTPC^TdTomato fluorescence (*Figure 4c*) and gene expression (*Figure 4—figure supplement 1c*). The SFTPC^TdTomato fluorescence could sustained up to Day-29 (2 weeks in alveolar maturation medium) (*Figure 4—figure supplement 1a,b*). As a control, we cultured Day-15 cardio-pulmonary progenitors on top of the transwell insert for air-liquid interface (ALI) culture, on 2D plastic surface for regular submerged culture or embedded in Growth Factor Reduced (GFR) Matrigel, and failed to detect obvious AT2 induction by Day-18 (*Figure 4c*, *Figure 4—figure supplement 2*, *Figure 4—figure supplement 3*). Consistent with the observations using fluorescence reporters, NKX2.1 and SFTPC gene expression was significantly upregulated in 3D suspension culture on Day-18 compared to the starting Day-15 cells or cells following ALI maturation (*Figure 4d and e*). This may in part be due to the reduction of cardiac progenitor as indicated by significant downregulation of NKX2.5 gene expression (*Figure 4f*, *Figure 4—figure supplement 1d*). Our results demonstrated 3D suspension culture as a robust platform to expedite alveolar maturation.

To elucidate how the co-induced cardiac lineage modulates the alveolar maturation process, we introduced activin A (20 ng/mL) during germ-layer specification (*Figure 4j*), which effectively inhibited mesoderm specification and led to pulmonary-only differentiation outcome on Day-15 (*Figure 3a and b*). In the absence of accompanying cardiac cells, although NKX2.1^GFP⁺ lung progenitors can be robustly induced and maintained, their alveolar maturation (as indicated by SFTPC^tdTomato reporter) following 3 days of maturation in 3D suspension culture was dramatically diminished compared to the cardio-pulmonary group (*Figure 4g–l*). Whole mount imaging of μTs on Day-18 showed pulmonary-only differentiation mainly comprised NKX2.1⁺ cells, while cardio-pulmonary μTs possessed a concentric arrangement of NKX2.1⁺ cells, surrounded by NKX2.5⁺ cells (*Figure 4i and j*). This was further supported by gene expression analysis of NKX2.1 (*Figure 4m*) and SFTPC (*Figure 4n*). Further extension of CKDCI maturation period for 2 weeks up to Day-29 in the pulmonary-only group failed to produce AT2 induction to a level comparable to that of the cardio-pulmonary group (*Figure 4—figure supplement 4*). This suggests that the cardiac lineage can serve as a cellular niche in supporting alveolar maturation.

We confirmed the reproducibility of our cardio-pulmonary co-induction protocol using another independent hiPSC line (BU1), including effective induction of endodermal and mesodermal mixture on Day-4 (*Figure 4—figure supplement 5a,b*), induction of pulmonary (NKX2.1) and cardiac (NKX2.5) progenitors on Day-15 (*Figure 4—figure supplement 5c,d*), and maturation of alveolar type two epithelium (Pro-SFTPC), and cardiac structural markers (cTnT and Sarcomeric Alpha Actinin) on Day-18 μTs (*Figure 4—figure supplement 5e*). Further, using BU1 without built-in fluorescence reporters, we

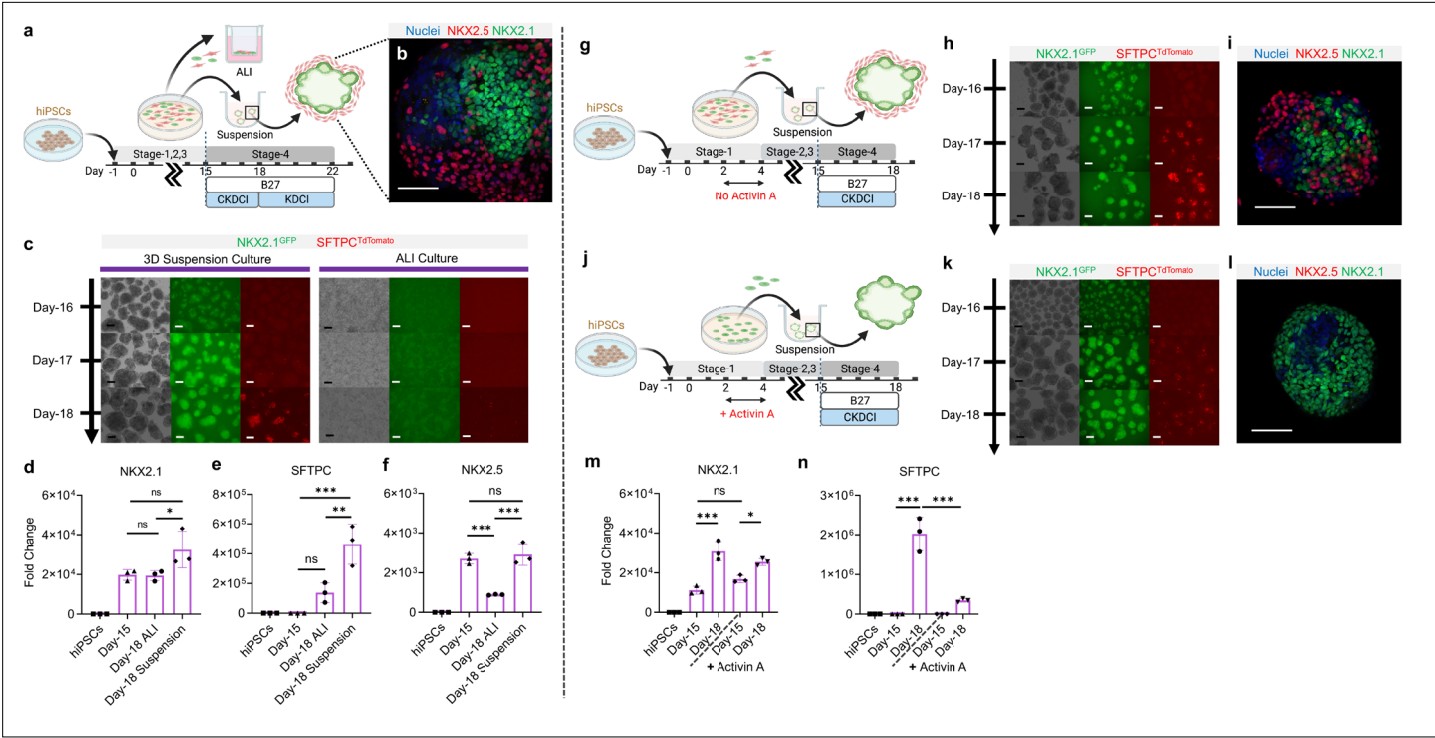

**Figure 4.** 3D suspension culture of cardio-pulmonary µTs expedites AT2 maturation. (**a**) Schematic diagram illustrating the Stage-4 maturation protocol involving replating of Day-15 cardiac and pulmonary progenitors onto ultra-low adhesion plate (for 3D suspension culture) or the transwell insert (for ALI culture). (**b**) Whole mount staining of cardiopulmonary µT on Day-18, scale bar 75 µm. (**c**) Live µT imaging of NKX2.1$^{GFP}$ and SFTPC$^{TdTomato}$ reporter signals during the first 3 days of maturation (Day-16 – Day-18). (**d-f**) qPCR analysis of hiPSC control, Day-15 cells and Day-18 cells (from ALI or suspension culture) for (**d**) NKX2.1 (n = 3 each; Day-15 vs. ALI, p = 0.9998; Day-15 vs. Suspension, p = 0.0547; ALI vs. Suspension, p = 0.0486), (**e**) SFTPC (n = 3 each; Day-15 vs. ALI, p = 0.1896; Day-15 vs. Suspension, p < 0.001; ALI vs. Suspension, p < 0.01). (**f**) NKX2.5 (n = 3 each; Day-15 vs. ALI, p < 0.001; Day-15 vs. Suspension, p = 0.8367; ALI vs. Suspension, p < 0.001). Scale bar = 125 µm. (**g–n**) Schematic diagram showing the differentiation procedure without (**g**) and with (**j**) Activin A during Stage-1 of differentiation (**h, k**) Live µT imaging of NKX2.1$^{GFP}$ and SFTPC$^{TdTomato}$ reporter signals during Day-16 to Day-18. Scale bar = 125 µm for 10 X images. (**i, l**) Whole mount staining of µTs on Day-18. (**m–n**) qPCR analysis of hiPSC control, Day-15 cells and Day-18 cells (from Activin-free or Activin) for (**m**) NKX2.1 (n = 3 each; Day-15 (No Activin) vs. Day-18 (No Activin), p < 0.001; Day-15 (Activin) vs Day-18 (Activin), p = 0.0147; Day-15 (No Activin) vs. Day-15 (Activin), p = 0.1316), (**n**) SFTPC (n = 3 each; Day-15 (No Activin) vs. Day-18 (No Activin), p < 0.001; Day-15 (Activin) vs Day-18 (Activin), p = 0.2417; Day-18 (No Activin) vs. Day-18 (Activin), p < 0.001). All data are mean ± SD. *p < 0.05; **p < 0.01; ***p < 0.001. Scale bar = 125 µm. 'n' refers to biological replicates. Diagram created using BioRender (http://biorender.com/).

The online version of this article includes the following source data and figure supplement(s) for figure 4:

**Source data 1.** Raw data for *Figure 4d, e, f, m and n*.

**Source data 2.** Raw data for *Figure 1c*.

**Figure supplement 1.** Co-maturation of Day-15 cardiac and pulmonary progenitors on ALI and 3D suspension culture platforms.

**Figure supplement 2.** Co-maturation of Day-15 cardiac and pulmonary progenitors on 2D submerged culture.

**Figure supplement 3.** Co-maturation of Day-15 cardiac and pulmonary progenitors in Matrigel Droplet and 3D suspension culture platforms.

**Figure supplement 3—source data 1.** Raw data for *Figure 4—figure supplement 3b*.

**Figure supplement 4.** Maturation of pulmonary progenitors derived from Activin A-based protocol on 3D suspension culture.

**Figure supplement 5.** Verification of cardio-pulmonary co-differentiation protocol on BU1 hiPSCs.

**Figure supplement 6.** Comparing fluorescence of NKX2.1$^{GFP}$ and SFTPC$^{TdTomato}$ of BU3-NGST vs. non-reporter BU1.

also confirmed that the GFP and TdTomato expression observed from BU3-NGST differentiation was not resulted from autofluorescence (*Figure 4—figure supplement 6*).

## Cardio-pulmonary segregation in the dual-lineage micro-tissue (MT)

Spatial-temporal regulation of WNT is crucial for early cardiac differentiation (*Lian et al., 2012*; *Zhao et al., 2019*; *Buikema et al., 2020*), however, continuous exposure to WNT activation is known to delay contractile maturation of cardiomyocytes (*Fan et al., 2018*). In parallel, exogenous WNT activation

using CHIR is essential for inducing AT2 maturation and its maintenance until the endogenous AT2 niche is established (*Jacob et al., 2017*; *Abdelwahab et al., 2019*; *Nabhan et al., 2018*; *Frank et al., 2016*). To investigate how CHIR removal regulates cardio-pulmonary maturation following AT2 establishment on Day-18 in 3D suspension culture (*Figure 4g*), we transitioned the maturation medium from CKDCI to KDCI without CHIR (*Figure 5a*; *Jacob et al., 2017*). To our surprise, upon CHIR removal, the cardiac and pulmonary components within each dual-lineage µT, which was initially arranged in the pulmonary-centered, concentric manner (*Figure 5b*), effectively reorganized over time and eventually segregated from each other (*Figure 5a and b*). When Day-15 cardio-pulmonary progenitors were transitioned to suspension culture directly in KDCI medium without CHIR, although they successfully underwent dual-lineage µT formation and segregation, there was no sign of AT2 maturation, echoing the importance of CHIR during AT2 induction (*Figure 5—figure supplement 1*; *Huang et al., 2013*; *Jacob et al., 2017*; *Nabhan et al., 2018*; *Frank et al., 2016*).

To quantitatively assess this segregation process, we performed time-lapse single-µT tracking and determined the percentage of overlap between the cardiac and pulmonary tissues by measuring the length of the overlapping border between the GFP+ and non-GFP components and normalizing it by the total perimeter of the GFP+ pulmonary component (*Figure 5c*). We compared the segregation process in the presence (CKDCI) and absence (KDCI) of CHIR, and found that although cardio-pulmonary segregation took place in both medium conditions, it was significantly expedited by the withdrawal of CHIR (*Figure 5d*,). To investigate the requirement of endogenous WNT signaling for this segregation process, we introduced inhibitors of canonical (IWP4) and non-canonical (NSC668036, a Dishevelled inhibitor) WNT signaling (*Li and Wang, 2018*), and did not detect any obvious difference in the segregation process as compared to the control KDCI condition (*Figure 5d*). In parallel with the cardio-pulmonary segregation, cardiac contraction was observed 7 days following CHIR withdrawal (*Video 1*). Immunohistochemical analysis demonstrated specific co-expression of NKX2.5 and cardiac troponin T (cTnT) in the segregated cardiac µT (*Figure 5e*).

## Cardio-pulmonary µT maturation

NKX2.1+ has the potential to differentiate into both proximal and distal airway epithelial cells. To characterize the lung epithelial composition in cardio-pulmonary µTs, we performed whole mount staining on Day-22 µTs. The induction of AT2 cells were further confirmed by the detection of the presence of lamellar bodies by transmission electron microscopy (*Figure 6a*) and by the positive immunofluorescence staining of pro-SFTPC+ (*Figure 6b*) and pro-SFTPB+ (*Figure 6—figure supplement 1a*). We also observed HOPX+ cells in the µTs (*Figure 6b*), and upregulated HOPX gene expression (*Figure 6—figure supplement 1b*) suggesting the presence of AT1-like cells. S100A4 staining indicated presence of mesenchyme in the µTs (*Figure 6—figure supplement 1c*), which may also play an important role in promoting alveologenesis *Hawkins et al., 2017*. In the meantime, these µTs did not express markers for proximal airway epithelium such as ciliated cells (FOXJ1) (*Figure 6—figure supplement 1d*), secretory cells (MUC5AC) (*Figure 6—figure supplement 1e*) and basal cells (p63) (*Figure 6—figure supplement 1f*), which can be readily observed in airway µT engineered from bronchial epithelial cells. In parallel, cardiac elements within the µTs exhibited the striated pattern as indicated by cTnT and Sarcomeric Alpha Actinin staining (*Figure 6c and d*). The cardiac contractile function was confirmed via the detection of calcium influx (*Video 2*), and its gradual reduction with increasing concentrations of Verapamil, a calcium channel blocker (*Figure 6e*).

## Discussion

Here, we described a novel strategy to model human cardio-pulmonary co-development using multi-lineage hiPSC differentiation. The current study primarily focused on co-induction of cardio-pulmonary progenitor cells, thus establishing the foundation for future investigations of how the crosstalk between these two organ lineages regulates their respective lineage maturation. We demonstrated that upon co-induction of mesoderm and endoderm, a series of shared signaling events were capable of driving simultaneous cardiac and pulmonary specification from their respective germ-layer progenitors. Upon transitioning the co-induced cardiac and pulmonary progenitors to 3D suspension culture, we observed expedited alveolar maturation within 3 days, which was supported by the accompanying cardiac lineage. In 3D suspension culture, each cardio-pulmonary µT effectively segregates into

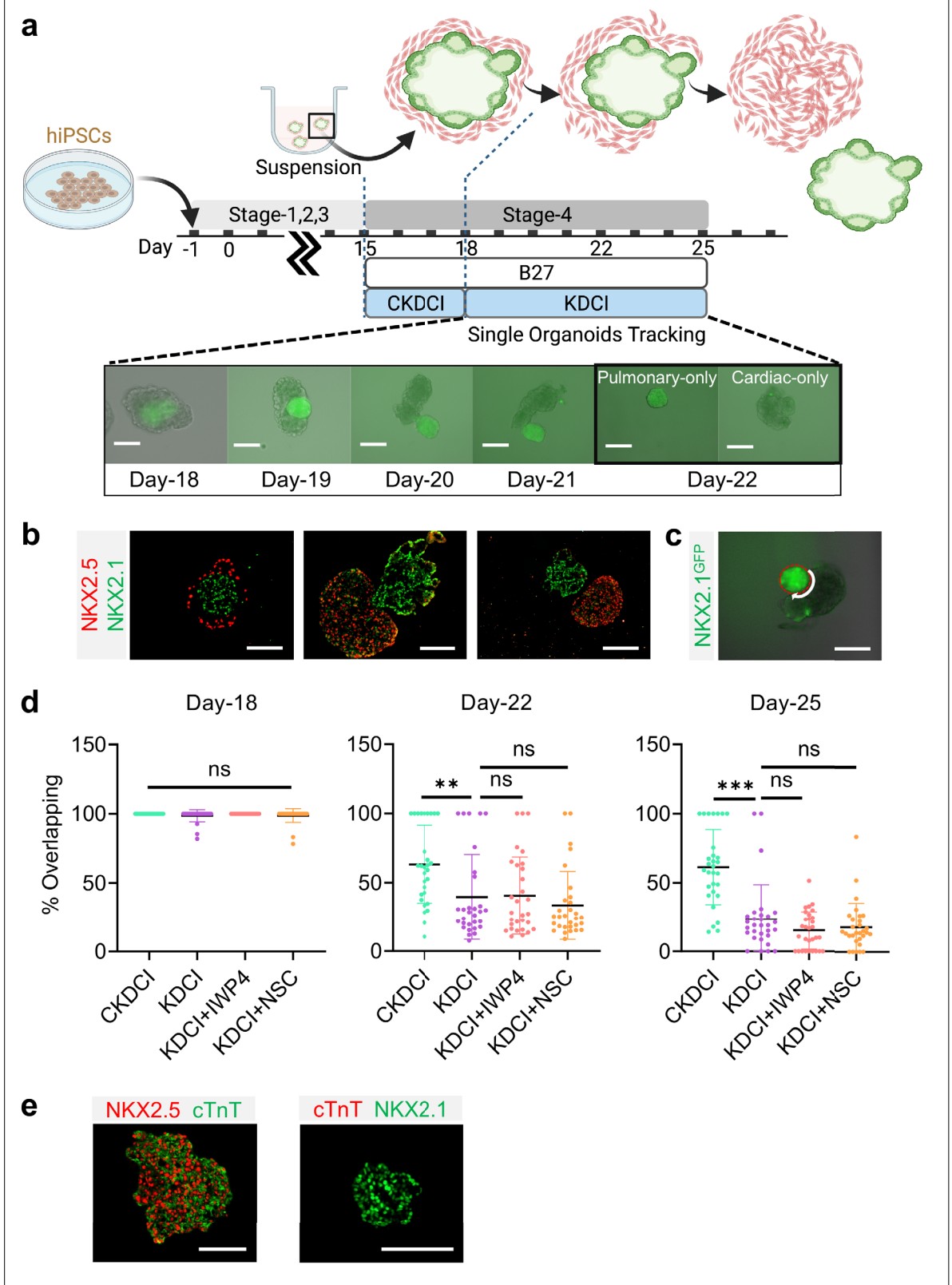

**Figure 5.** Cardio-pulmonary segregation in the dual-lineage μT. (**a**) Schematic diagram illustrating the timeline for the investigation. Scale bar = 125 μm (**b**) Histological analysis of cardio-pulmonary μTs at different stages of segregation. Scale bar = 125 μm (**c**) Diagram showing measurement of the total perimeter of GFP+ pulmonary compartment (red color) and its overlapping perimeter with non-GFP compartment (white color) using Image J. Scale bar = 125 μm (**d**) Box plot showing percentage overlapping region of GFP+ with non-GFP tissues on Day- 18 (n = 30 each;; CKDCI vs. KDCI, p = 0.3979;

*Figure 5 continued on next page*

*Figure 5 continued*

CKDCI vs. KDCI+ IWP4; p > 0.9999; CKDCI vs. KDCI+ NSC, p = 0.4293; KDCI vs. KDCI+ IWP4, p = 0.3979; KDCI vs. KDCI+ NSC, p > 0.9999; KDCI+ IWP4 vs. KDCI+ NSC, p = 0.4293), Day-22 (n = 30 each; CKDCI vs. KDCI, p = 0.0077; CKDCI vs. KDCI+ IWP4; p = 0.0112; CKDCI vs. KDCI+ NSC, p < 0.001; KDCI vs. KDCI+ IWP4, p = 0.9994; KDCI vs. KDCI+ NSC, p = 0.8318; KDCI+ IWP4 vs. KDCI+ NSC, p = 0.7674) and Day-25 (n = 30 each; CKDCI vs. KDCI, p < 0.001; CKDCI vs. KDCI+ IWP4; p < 0.001; CKDCI vs. KDCI+ NSC, p < 0.001; KDCI vs. KDCI+ IWP4, p = 0.4271; KDCI vs. KDCI+ NSC, p = 0.7275; KDCI+ IWP4 vs. KDCI+ NSC, p = 0.9623). (**e**) Histological analysis of cTnT expression on the segregated cardiac and pulmonary µTs, with co-staining of NKX2.5 and NKX2.1. Scale bar = 125 µm. All data are mean ± SD. *p < 0.05; **p < 0.01; ***p < 0.001. 'n' refers to biological replicates. Diagram created using BioRender (http://biorender.com/).

The online version of this article includes the following source data and figure supplement(s) for figure 5:

**Source data 1.** Raw data for *Figure 5d*.

**Figure supplement 1.** Day-15 cells in CKDCI vs. KDCI for co-maturation.

**Figure supplement 1—source data 1.** Raw data for *Figure 5—figure supplement 1c,e*.

---

separate cardiac and pulmonary µTs, which was partially inhibited by WNT activation. This study therefore delivers an effective in vitro model for studying the mechanistic interplay between the developing heart and lung during human embryogenesis.

The extensive cardio-pulmonary mutual interaction during organogenesis has been well documented in the mouse model (*Peng et al., 2013*; *Hoffmann et al., 2009*; *Steimle, 2018*); however, the translatability of these findings to human embryogenesis remains elusive due to the lack of proper model systems. Human pluripotent stem cell differentiation has offered an effective means for recapitulating and investigating human organogenesis, and tremendous progress has been made toward directed cardiac or pulmonary specification (*Lian et al., 2012*; *Mummery et al., 2012*; *Burridge et al., 2014*; *Lian et al., 2015*; *Lee et al., 2017*; *Kattman et al., 2011*; *Chen et al., 2017*; *Huang et al., 2013*; *Jacob et al., 2017*; *Dye et al., 2015*; *Gotoh et al., 2014*; *Wong et al., 2012*; *Laflamme et al., 2007*; *D'Amour et al., 2005*; *Longmire et al., 2012*; *Lian et al., 2013*; *Mou et al., 2012*; *Green et al., 2011*). However, almost all existing models have been focusing on one parenchymal lineage at a time, and therefore lack the ability to support the investigation of inter-organ crosstalk. Here, building on the established understanding of signaling events necessary for cardiac and pulmonary induction (*Lian et al., 2012*; *Mummery et al., 2012*; *Burridge et al., 2014*; *Lian et al., 2015*; *Lee et al., 2017*; *Kattman et al., 2011*; *Chen et al., 2017*; *Huang et al., 2013*; *Jacob et al., 2017*; *Dye et al., 2015*; *Gotoh et al., 2014*; *Wong et al., 2012*; *Laflamme et al., 2007*; *D'Amour et al., 2005*; *Longmire et al., 2012*; *Lian et al., 2013*; *Mou et al., 2012*; *Green et al., 2011*), we have developed a robust protocol for the simultaneous co-differentiation of cardiac and pulmonary lineages from hiPSCs. Within our co-differentiation system, unrestricted interaction between cells of both lineages is enabled even before their lineage commitment.

Most current attempts on pulmonary induction from hiPSCs relies on initial nodal activation using growth factor (Activin A) supplementation, which is critical for definitive endoderm specification. Here, we showed that by fine tuning WNT signaling using a small-molecule inhibitor of GSK-3β (CHIR), robust induction of endoderm and subsequently lung progenitors can be achieved without any exogenous growth factors. This is consistent with the observation that CHIR was capable of inducing cardiac differentiation in replacement of combined effect of exogenous Activin A and BMP4 (*Lian et al., 2012*). In addition, *Martyn et al., 2019* demonstrated that WNT is sufficient to induce primitive streak, which then creates a gradient of signals (BMP, WNT, Nodal) that could further specify fate towards mesoderm and endoderm lineages (*Martyn et al., 2019*). Nonetheless, Nodal and BMP signaling remains crucial in mesoderm and endoderm specification, as inhibition of these signals effectively terminated cardio-pulmonary co-induction.

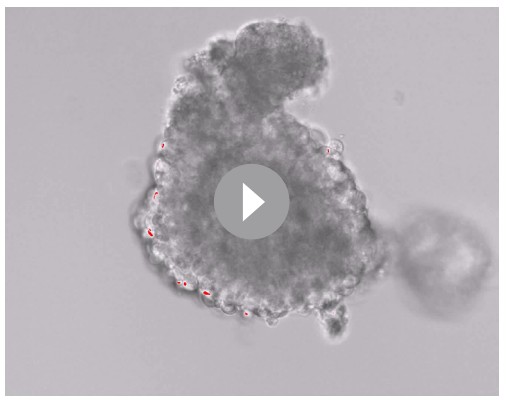

**Video 1.** Contacting cardiac µT following 7 days after withdrawal of CHIR.
https://elifesciences.org/articles/67872/figures#video1

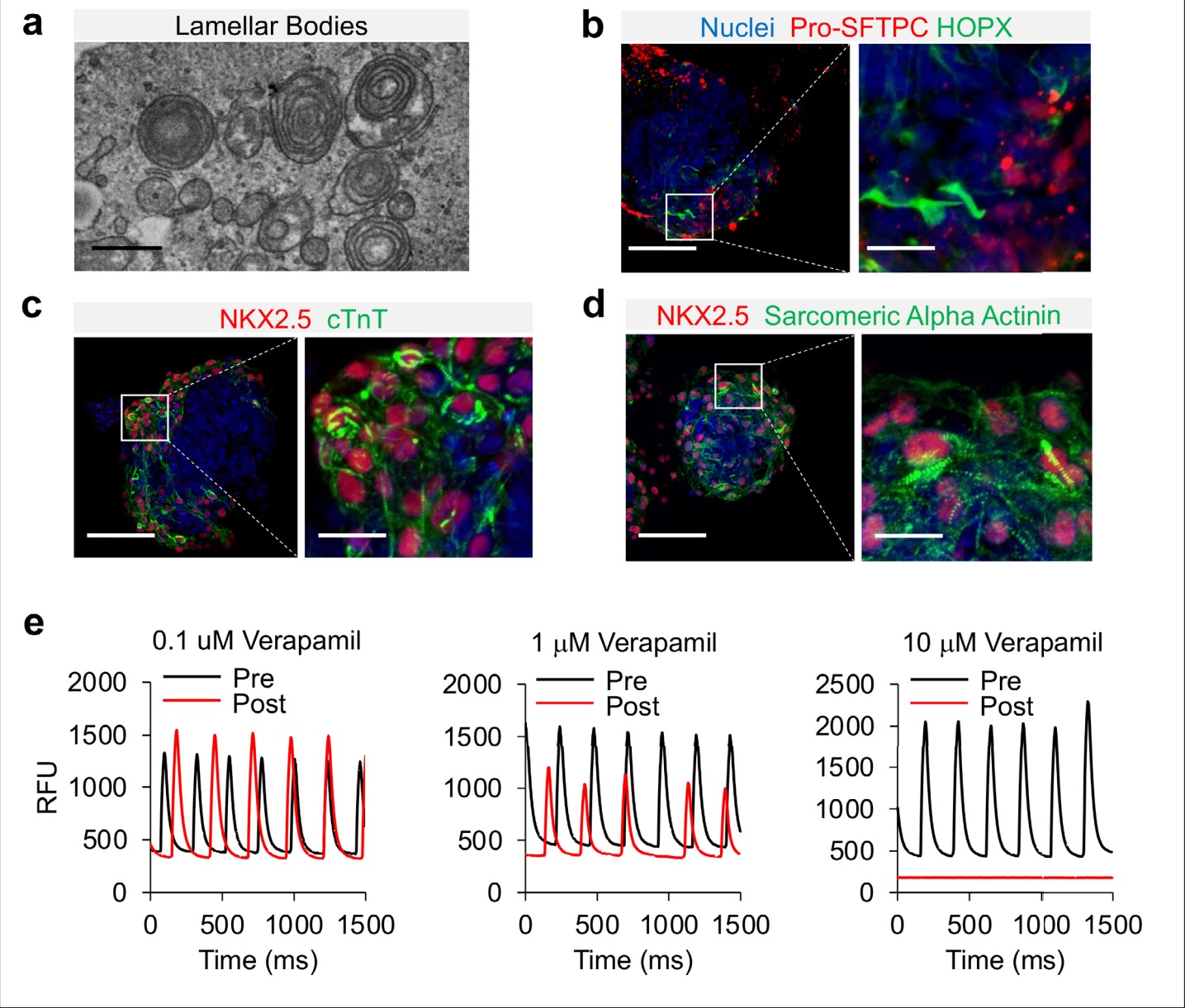

**Figure 6.** Characterization of cardio-pulmonary µT maturation. (**a**) Lamellar bodies found in cardio-pulmonary µT. Scale bar = 1 µm. Cardio-pulmonary µT expressing (**b**) Pro-SFTPC and HOPX. Cardio-pulmonary µT also exhibited striated pattern as indicated by (**c**) cTnT and (**d**) Sarcomeric Alpha Actinin. Scale bar = 125 µm. (**e**) Calcium imaging of cardiac µTs following treatment with Verapamil.

The online version of this article includes the following source data and figure supplement(s) for figure 6:

**Source data 1.** Raw data for *Figure 6e*.

**Figure supplement 1.** Characterization of cardio-pulmonary µT.

**Figure supplement 1—source data 1.** Raw data for *Figure 6—figure supplement 1b*.

Our study demonstrated the need for endogenous TGF-β signaling for effective cardio-pulmonary induction, as well as the critical role of endogenous BMP signaling in cardiogenesis. Furthermore, we found that temporal-specific action of the same set of small molecules regulating TGF-β and WNT signaling was capable of simultaneously driving mesoderm-to-cardiac and endoderm-to-pulmonary specification. Moreover, BMP4 has been shown to improve NKX2.1[+] lung progenitor specification in both mouse and hiPSCs (*Huang et al., 2013*; *Jacob et al., 2017*; *Serra et al., 2017*). In our system, endogenous instead of exogenous BMP signaling was required during a developmental

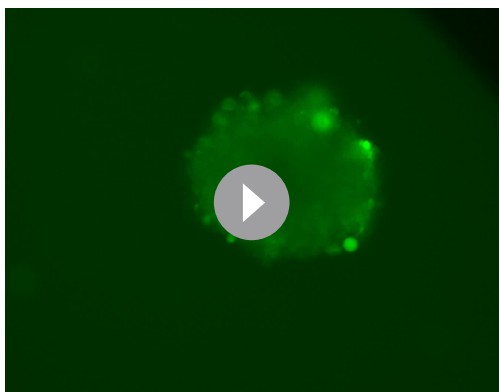

**Video 2.** Calcium influx capability of cardiac μT loaded with Cal-520.

https://elifesciences.org/articles/67872/figures#video2

stage corresponding to foregut ventralization for effective co-emergence of cardiac and pulmonary progenitors. This is in line with the close spatial positioning of the developing heart and lung primordia within embryonic body patterning, which implies their exposure to a similar paracrine microenvironment (*Steimle, 2018*; *Herriges and Morrisey, 2014*).

To achieve alveolarization, NKX2.1⁺ lung progenitors are usually embedded in extracellular matrices, such as Matrigel and collagen (*Jacob et al., 2017*; *Dye et al., 2015*; *de Carvalho et al., 2019*). Here, we established an effective approach that enabled AT2 cell maturation within 3 days in suspension culture of 3D cell aggregates spontaneously formed from Day-15 cardiac and pulmonary progenitors. We further demonstrated that the presence of accompanying cardiac lineage is critical for robust alveolar induction. This observation is consistent with the recently reported inter-dependence between cardiac and pulmonary lineages during embryogenesis (*Steimle, 2018*). In the absence of cardiac lineage, NKX2.1⁺ lung progenitors could not achieve as effective AT2 maturation as what is observed in cardio-pulmonary μTs. This is again emphasizing that chemical cues derived from heart field also plays a role in lung development (*Peng et al., 2013*; *Steimle, 2018*). However, it should be noted that pulmonary mesenchyme could also augment distal lung differentiation, which requires further investigation with the ability to differentiate the participation and contribution of each mesodermal cell lineages (*Hawkins et al., 2017*; *McCulley et al., 2015*; *Leeman et al., 2019*). In addition, the presence of mesoderm-derived stromal cells have also been shown to be essential for effective alveolarization in vivo (*Herriges and Morrisey, 2014*; *Domyan et al., 2011*; *Rankin et al., 2016*) and in vitro (*Gotoh et al., 2014*; *Hawkins et al., 2017*). Furthermore, cells of the mesodermal lineage are known to be robust producers of extracellular matrix, which may also contribute to the effective alveolar maturation in the absence of external extracellular matrix support. The ability to enable effective alveolar induction from hiPSC-derived lung progenitors with the convenient of suspension culture also opens the door to large-scale production of alveolar cells, a critical challenge in regenerative medicine applications.

Using dual-lineage cardio-pulmonary μTs formed from the co-induced progenitors, we observed a novel process of cardio-pulmonary tissue segregation. The human body cavities are highly crowded spaces, filled with different tissues and organs that are in close contact with each other. It remains enigmatic how inter-organ boundaries are maintained to prevent undesired cell migration or tissue merging. Our cardio-pulmonary tissue segregation model suggests an intrinsic mechanism that effectively establishes a boundary between two distinct parenchymal lineages even when they are initially mingled together. Although no model of collective migration has been described in the context of cardio-pulmonary development, studies in other model systems suggest cell-cell communication and paracrine signaling (e.g. WNT) to be crucial for directed cell migration during development (*Ciruna and Rossant, 2001*; *De Calisto et al., 2005*; *Carmona-Fontaine et al., 2008*). Here, we found that exogenous WNT activation via GSK-3β inhibition effectively slowed down the cardio-pulmonary segregation, while inhibition of endogenous WNT (canonical and non-canonical) did not obviously affect the process. In consistence with our observation, it has been shown that the inhibition of non-canonical WNT signaling does not stop collective cell migration but distorting migration direction (*Li and Wang, 2018*).

Despite the presence of AT1-like cells in cardio-pulmonary μTs based on HOPX staining (*Wang et al., 2018*; *Frank et al., 2019*; *Liebler et al., 2016*), the organization of these cells remained unlike the thin, flat squamous cells found in the native lung (*Yang et al., 2016*; *Williams, 2003*). Although HOPX expression has been observed in cardiomyocytes (*Friedman et al., 2018*), we did not observe HOPX expression following establishment of cardio-pulmonary progenitors, and the induction of HOPX coincided with alveolar maturation during Day-15 to Day-18. Further study is needed to explore

the possibility of obtaining AT1-like cells with a more mature phenotype using multiple markers. Our work demonstrates early emergence of SFTPC-expressing cells as early as Day-18, with CHIR and 3D suspension culture being key driving forces that promote alveologenesis (*Huang et al., 2013*; *Jacob et al., 2017*), Removal of CHIR on Day-18 led to a decreased in SFTPC expression on Day-22 (*Figure 4—figure supplement 1c*), suggesting that the complete alveolar niche was not fully established yet. Future investigation is needed to further clarify components of the supporting alveolar niche as well as its timeline of establishment. In parallel, cardiac contractility of the µTs was initiated 7 days following removal of CHIR, consistent with previous studies showing that GSK-3β inhibition promotes cardiomyocyte proliferation but hinders contraction by affecting myofibrillar architecture (*Buikema et al., 2020*; *Wang et al., 2016*; *Tseng et al., 2006*).

Heart begins to form around gestation week 3, which is the earliest organ to be developed during embryogenesis (*Buckingham et al., 2005*; *Tan and Lewandowski, 2020*). Lung development initiates soon afterwards during weeks 4–7 (*Schittny, 2017*). In the present study, we showed that the progenitor cells of both heart and lung can be simultaneously induced following 15 days of co-differentiation from hiPSCs. More comprehensive time-series analysis will be necessary to further delineate the fine temporal relationship between heart and lung lineage specification, which is expected to provide fundamental insights regarding cardio-pulmonary crosstalk during their paralleled organogenesis.

In conclusion, our work focuses on specification of cardio-pulmonary progenitors that have the potential to further mature into their respective descendent lineages. Further morphological and functional maturation of both cardiac and pulmonary lineages, as well as their crosstalk during this process will require future in-depth investigation. In addition, our work offers a novel model for investigating the molecular and cellular mechanisms underlying human cardio-pulmonary co-development and tissue boundary formation. We also expect this work to be of potential use for studying congenital diseases affecting both cardiovascular and pulmonary systems, such as congenital diaphragmatic hernia.

## Materials and methods

### Materials

Detailed information regarding reagents for culture and differentiation medium was summarized in *Supplementary file 1*. Reagents, equipment, and probes for quantitative PCR (qPCR) analysis antibodies and reagents for immunofluorescence staining were summarized in Key Resources Table.

### Maintenance of human-induced pluripotent stem cells (HiPSCs)

The BU3-NGST and BU1 hiPSC lines were obtained as kind gifts from the laboratories of Dr. Darrell Kotton and Dr. Finn Hawkins (Boston University). BU3 hiPSC line was derived from a healthy donor and carries both NKX2.1$^{GFP}$ (NG) and Surfactant protein C (SFTPC)$^{tdTomato}$ (ST) reporters (*Jacob et al., 2017*; *Hawkins et al., 2017*). BU1 hiPSC line was also derived from healthy donor but without any reporters. hiPSCs were maintained on Matrigel-coated (ESC-qualified) six-well tissue culture plate with mTESR1 Plus medium with regular medium changed every other day. hiPSCs passaging was performed every 5–7 days using ReLESR at a plating ratio of 1:10. All cells used in this study were tested negative for mycoplasma contamination using Universal Mycoplasma Detection Kit (ATCC, 30–1012 K).

Simultaneous induction of cardiac and pulmonary progenitors from hiPSCs hiPSCs maintained in mTESR Plus were dissociated into single cells using StemPro Accutase. 150,000 cells/cm$^2$ on hESC-qualified Matrigel-coated 96-well plate, and cultured in mTESR Plus supplemented with 10 µM Y-27632 (ROCK inhibitor) for 24 hr prior to differentiation. The overall protocol for stepwise cardio-pulmonary co-differentiation was summarized in *Supplementary file 1*. To induce a balanced mixture of mesodermal and definitive endodermal cells, hiPSCs were first incubated in mTESR Plus medium supplemented with different concentration (4, 7, 10 µM) of CHIR99021 (GSK3β inhibitor) and 10 µM Y-27632 for 48 hr. This was followed by an additional 48 hr incubation in serum-free differentiation medium consisting of RPMI 1640 supplemented with 2% B-27 minus insulin, 1 x GlutaMAX and 10 µM Y-27632. In some experiments, Activin A (20 ng/mL), BMP4 (20 ng/mL), A8301 (Nodal or TGF-β inhibitor, 1 µM) or DMH-1 (BMP inhibitor, 2 µM) were introduced to examine how Nodal and BMP signaling regulated mesodermal and endodermal specification. Differentiation outcomes were assessed by immunostaining and qPCR analysis of mesodermal (NCAM1) and definitive endodermal (SOX17) markers.

Following Stage-1, all subsequent differentiation procedures were performed using medium recipes formulated based on RPMI 1640 medium supplemented with 2% B-27 and 1 x GlutaMAX, referred to as 'basal medium'. To initiate simultaneous cardiac and pulmonary specification, Day-4 cells were incubated for 4 days in Stage-2 medium, containing basal medium supplemented with 1 μM A8301, 5 μM IWP4 and 10 μM Y-27632. In some experiments, co-differentiation medium without either A8301 or IWP4 was utilized to investigate the impact of inhibition of TGF-β and WNT signaling.

Following Stage-2, to induce simultaneous specification of both cardiac and lung progenitors, co-differentiating cells were incubated for 7 days in Stage-3 medium containing basal medium supplemented with 3 μM CHIR99021 and 100 nM Retinoic acid (RA). Green fluorescence of the NKX2.1[GFP] reporter was examined daily using EVOS FL Auto 2 Imaging System to monitor the emergence of lung progenitors. On Day-15 of co-differentiation, the expression of cardiac (NKX2.5) and lung (NKX2.1) progenitor markers was evaluated by immunofluorescence staining and qPCR.

## Co-maturation of cardio-pulmonary progenitors in air-liquid interface (ALI) culture

On Day-15 of cardio-pulmonary co-differentiation, cells were dissociated into single cells using TrypLE Express, and re-plated at 500,000 cells/cm² onto the apical side of each 24-well Transwell insert (pore size of 0.4 μm, pre-coated with 1% growth factor-reduced Matrigel) in 100 μL maturation medium. Basolateral side of the transwell insert was filled with 500 μL of maturation medium. The maturation medium was basal medium supplemented with 3 μM **C**HIR99021, 10 ng/mL Keratinocyte growth factor (**K**GF), 50 nM **D**examethasone, 0.1 mM 8-bromoadenosine 3', 5'-cyclic monophosphate (**c**AMP, AMP-activated protein kinase activator) and 0.1 mM 3-**i**sobutyl-1-methylxanthine (IBMX, PKA activator), which was referred to as **CKDCI** medium. 10 μM Y-27632 was added during the initial 24 hrs following re-plating. The next day, all medium on the apical side was removed. 200 μL of fresh CKDCI medium without Y-27632 was added to the basolateral side to establish ALI culture, and was replaced daily. Red fluorescence from the SFTPC[TdTomato] reporter was examined daily using EVOS Imaging System to monitor the emergence of alveolar type 2 (AT2) cells. On Day-3 of ALI maturation, Transwell membrane were excised from the insert, and analyzed by qPCR (NKX2.1, SFTPC).

## Co-maturation of cardio-pulmonary μTs in 3D suspension culture

On Day-15 of cardio-pulmonary co-differentiation, cells were dissociated into single cells using TrypLE Express. A total of 250,000 cells in 500 μL CKDCI maturation medium was transferred into each well of 24-well ultra-low adherence plate and cultured with agitation at 125 rpm to form cardio-pulmonary μTs. 10 μM Y-27632 was added during the initial 24 hr following re-plating. Following 3 days of culture in CKDCI medium, CHIR99021 was removed and μT culture was continued in KDCI medium for an additional 7 days. At desired time points of 3D suspension maturation, μTs were analyzed by histology (NKX2.1, NKX2.5, cTnT) and qPCR analysis (NKX2.1, SFTPC).

## Embedding cardio-pulmonary μTs in matrigel droplet

On Day-15 of cardio-pulmonary co-differentiation, cells were dissociated into single cells using TrypLE Express. A total of 5000 cells in 50 μL Growth Factor Reduced (GFR) Matrigel were dropped on 24-well plate and cultured in CKDCI maturation medium. 10 μM Y-27632 was added during the initial 24 hr following re-plating. Following 3 days of culture, the Matrigel was dissolved in ice-cold EDTA, and the embedded cells were then recovered for RNA extraction and qPCR analysis of NKX2.1, SFTPC, NKX2.5.

## Airway μTs

To generate airway μTs, 96-well plate was coated with 50 μL of 40% (v/v) GFR Matrigel diluted in PneumaCult-ALI Maintenance Medium. The normal human bronchial epithelia were resuspended in 40% (v/v) GFR Matrigel in PneumaCult-ALI Maintenance Medium and added to the coated wells. A total of 100 μL PneumaCult-ALI Maintenance Medium was placed in the wells and changed every other day. Airway μTs formed were harvested following over 3 weeks of differentiation and fixed for immunostaining.

## Single μT time-lapse imaging and analysis

To investigate the segregation of cardio-pulmonary μTs into their respective cardiac and lung μTs, following 3 days suspension culture in CKDCI medium in 24-well ultra-low adherence plate, single

µT was transferred into each well in 96-well ultra-low adherence plate and cultured for an additional 7 days. The following medium recipes were examined for cardio-pulmonary segregation: KDCI medium, KDCI medium supplemented with 3 µM CHIR99021, KDCI medium with 5 µM IWP4, and KDCI medium with 50 µM NSC668036. Time-lapse imaging was performed on Day- 18, Day-22, and Day-25 following µT transfer to monitor the segregation process. The pulmonary compartment within each cardio-pulmonary µT was tracked based on the NKX2.1$^{GFP}$ reporter. To quantify the segregation between the two compartments within each µT. Image J was used to measure the overlapping perimeter between GFP$^+$ (pulmonary) and non-GFP (cardiac) compartments, which was then normalized to total perimeter of GFP$^+$ compartments and expressed as the percentage of overlapping.

$$Percent\ overlapping\ (\%) = \frac{Overlapping\ Perimeter\ of\ GFP\ and\ nonGFP\ compartments\ (\mu m)}{Total\ Perimeter\ of\ GFP\ organoids\ (\mu m)} \times 100\%$$

## qPCR analysis

Total RNA was extracted using TRIzol, processed by chloroform extraction, precipitated using 1 volume of absolute isopropanol with 50 µg/mL of RNase-free glycoblue as carrier, washed with 75% ethanol, air-dried, solubilized in RNase-free water and quantified using NanoDrop 2000 spectrophotometer. cDNA was synthesized via reverse transcription of 1 µg total RNA with random hexamers and the High-Capacity cDNA Reverse Transcription kit according to manufacturer's instruction. Real-time qPCR analysis was performed on CFX96 Touch Real-Time PCR Detection System using TaqMan probes. Each reaction mixture was prepared by combining 1 µL of probe, 10 µL of TaqMan Master Mix, 1 µL of cDNA (equivalent to 50 ng), and the final volume was brought up to 20 µL. The final Ct value was normalized to housekeeping gene (β-actin), using comparative Ct method. Unless otherwise specified, baseline, defined as fold change = 1, was set as undifferentiated hiPSCs, or if undetected, a cycle number of 40 was assigned to allow fold change calculations (*Jacob et al., 2017*). List of TaqMan probes was summarized in Key Resources Table.

## Immunofluorescence staining on 2D cell samples

Cells were fixed with ice-cold methanol, air-dried, rehydrated with phosphate-buffered saline (PBS), permeabilized with 1% (v/v) Triton X-100, blocked in 1% (w/v) bovine serum albumin in PBS (blocking buffer), incubated with primary antibodies diluted in blocking buffer at 4 °C overnight, and incubated with corresponding fluorescence-conjugated secondary antibodies in blocking buffer at room temperature (RT) for 45 min. Nuclear counterstain was performed using Hoechst-33342 (1:500) in PBS. Fluorescence images were acquired using EVOS Imaging System. All antibodies used and their respective dilution were summarized in Key Resources Table.

## Histology

The µTs were fixed with 4% paraformaldehyde, embedded in HistoGel and then in paraffin. Tissue processing and paraffin embedding was performed in Research Histology Lab of Pitt Biospecimen Core at the University of Pittsburgh Medical Center (UPMC) Shadyside Hospital. Paraffin blocks were sectioned at 5 µm thickness, transferred onto glass slides, rehydrated by sequential incubation in Histoclear, 100% ethanol, 95% ethanol and distilled water. To unmask antigen, slides were treated with Antigen Unmasking Solution at 95 °C for 20 min and cooled down to RT. Immunofluorescence staining was performed as described above for 2D cell samples. After the final wash, slides were mounted with DAPI Fluoromount-G, and imaged using EVOS Imaging System. All antibodies used and their respective dilution were summarized in Key Resources Table.

## Flow cytometry

Cells were dissociated into single cells via incubation with TrypLE for 15 min. For NKX2.1$^{GFP}$ assessment, approximately $3 \times 10^5$ cells were resuspended in FACS buffer (DPBS with 1% FBS) and incubated with DAPI for 10 min on ice, followed by three washes prior to analysis. For indirect labeling of NKX2.5, cells were first trypsinized and stained with Fixable Violet Dead Cell Stain Kit (Thermo Fisher Scientific) for 10 min on ice. Cells were then fixed with 4% PFA on ice for 20 min, followed by three times washes with 1% BSA in PBS. Cells were permeabilized with 1% Triton X-100 for 20 min, followed by blocking for 30 min prior to adding primary antibody for overnight incubation. Next day, cells were washed three times in 1% BSA in PBS and incubated with fluorophore-conjugated secondary antibody

for 1 hr. Following three washes with 1% BSA in PBS, cells were re-suspended in FACS buffer for flow cytometry analysis at Unified Flow Core of Department of Immunology at University of Pittsburgh Medical Center.

## Contraction and calcium signal

To assess contraction of cardiac µT, segregated cardiac µT was stained with 5 µM of Cal-520 AM (AAT Bioquest, 21130), a calcium indicator dye. The concentration-response of cardiac µTs to calcium channel blocker (Verapamil) were assessed by treating the µTs with 0.1, 1, and 10 µM of Verapamil for 10 min. Calcium imaging (500 frames per second) was performed pre- and post- Verapamil treatment using a Prime 95B Scientific CMOS camera (Photometrics) mounted on an epifluorescent stereomicroscope (Nikon SMZ1000) with a GFP filter and an X-cite Lamp (Excelitas).

## TEM

Cardio-pulmonary µTs were fixed in 2.5% glutaraldehyde in 0.1 M PBS (pH7.4) for at least 1 hr. After three washes in 0.1 M PBS for 10 min each, the µTs were post fixed in 1% Osmium tetroxide containing 1% potassium ferricyanide at 4 °C for 1 hr, followed by three washes in 0.1 M PBS for 10 min each. µTs were dehydrated in graded series of ethanol starting from 30%, 50%, 70%, 90% and finally 100% of ethanol for 10 min each. µTs were further dehydrated epon for 1 hr at RT. This step was repeated for another three times prior to embedding in pure epon at 37 °C for 24 hr. Finally, the µTs were cured for 48 hr at 60 °C. The presence of lamellar body in cardio-pulmonary µTs were identified using JEM 1400 Flash TEM.

## Statistics

Statistical methods relevant to each figure were outlined in the accompanying figure legend. At least three biological replicates were performed for each group under comparison. Unless otherwise indicated, unpaired, two-tailed Student's t tests were applied to comparisons between two groups. For comparisons among three or more groups, one-way ANOVA was performed followed by Tukey multiple comparison tests. Results are displayed as mean ± SD, with $p < 0.05$ considered statically significant. n values referred to biologically independent replicates.

# Acknowledgements

This work was supported by Samuel & Emma Winters Foundation A025662 (to XR) and the Department of Biomedical Engineering and College of Engineering at Carnegie Mellon University. We are grateful to Drs. Yu-li Wang and David Li for advice on collective cell migration, to Misti West for laboratory management, and to Anthony Green and the Pitt Biospecimen Core at the University of Pittsburgh for assistance with histology. We also thank Barbie Varghese for proofreading the manuscript.

# Additional information

### Competing interests

Wai Hoe Ng, Elizabeth K Johnston, Xi Ren: This author is a co-inventor of a related provisional patent application (No. 63/124422) entitled 'Methods for simultaneous cardio-pulmonary differentiation and alveolar maturation from human pluripotent stem cells'. The other authors declare that no competing interests exist.

### Funding

| Funder | Grant reference number | Author |
| --- | --- | --- |
| Samuel & Emma Winters Foundation | A025662 | Xi Ren |
| Carnegie Mellon University | | Xi Ren |

The funders had no role in study design, data collection and interpretation, or the decision to submit the work for publication.

## Author contributions
Wai Hoe Ng, Conceptualization, Data curation, Formal analysis, Investigation, Methodology, Validation, Visualization, Writing – original draft, Writing – review and editing; Elizabeth K Johnston, Investigation, Methodology, Writing – review and editing; Jun Jie Tan, Supervision, Writing – review and editing; Jacqueline M Bliley, Investigation, Writing – review and editing; Adam W Feinberg, Finn Hawkins, Resources, Supervision, Writing – review and editing; Donna B Stolz, Ming Sun, Investigation; Piyumi Wijesekara, Investigation, Visualization; Darrell N Kotton, Resources, Supervision; Xi Ren, Conceptualization, Data curation, Formal analysis, Funding acquisition, Investigation, Project administration, Resources, Supervision, Writing – original draft, Writing – review and editing

## Author ORCIDs
Wai Hoe Ng (ID) http://orcid.org/0000-0002-6876-6542
Darrell N Kotton (ID) http://orcid.org/0000-0002-9604-8476
Xi Ren (ID) http://orcid.org/0000-0003-3187-1311

## Decision letter and Author response
Decision letter https://doi.org/10.7554/eLife.67872.sa1
Author response https://doi.org/10.7554/eLife.67872.sa2

---

# Additional files

## Supplementary files
- Supplementary file 1. Media Recipes/Composition.
- Transparent reporting form

## Data availability
All data supporting the findings of this study are available within the article and its supplementary files. Source data files have been provided for Figures 1 to 6.

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

# Appendix 1

### Appendix 1—key resources table

| Reagent type (species) or resource | Designation | Source or reference | Identifiers | Additional information |
|---|---|---|---|---|
| Gene (*Homo sapiens*) | β-actin | GenBank (NM_001101.3) | Hs01060665_g1 | |
| Gene (*Homo sapiens*) | NKX2.1 | GenBank (NM_001079668.2) | Hs00968940_m1 | |
| Gene (*Homo sapiens*) | FOXA2 | GenBank (NM_02178.4) | Hs00232764_m1 | |
| Gene (*Homo sapiens*) | SOX17 | GenBank (NM_022454.3) | Hs00751752_s1 | |
| Gene (*Homo sapiens*) | NKX2.5 | GenBank (NM_004387.3) | Hs00231763_m1 | |
| Gene (*Homo sapiens*) | SFTPC | GenBank (NM_001172357.1) | Hs00161628_m1 | |
| Gene (*Homo sapiens*) | NCAM1 | GenBank (NM_000615.6) | Hs00941821_m1 | |
| Gene (*Homo sapiens*) | HOPX | GenBank (NM_001145459.1) | Hs04188695_m1 | |
| Cell line (*Homo sapiens*) | BU3-NGST | Boston University (Kotton's Lab) | RRID:CVCL_WN82 | |
| Cell line (*Homo sapiens*) | BU1 | Boston University (Kotton's Lab) | - | |
| Cell line (*Homo sapiens*) | Normal Human Bronchial Epithelial (NHBE) cells | Lonza | Cat# CC-2541; RRID:CVCL_S124 | |
| Antibody | anti-NKX2.1 (Rabbit monoclonal) | Abcam | Cat# ab76013; RRID:AB_1310784 | 1:500 |
| Antibody | anti-NKX2.5 (Goat polyclonal) | R&D Systems | Cat# AF2444; RRID:AB_355269 | 1:500 |
| Antibody | anti-Pro-SFTPB (Rabbit polyclonal) | Seven Hills | Cat# WRAB-55522 RRID:AB_ | 1:200 |
| Antibody | anti-Pro-SFTPC (Rabbit polyclonal) | Seven Hills | Cat# WRAB-9937; RRID:AB_451721 | 1:500 |
| Antibody | anti-HOPX (Mouse monoclonal) | Santa Cruz | Cat# Sc-398703; RRID:AB_2687966 | 1:100 |
| Antibody | anti-cTnT (Mouse monoclonal) | Thermo Fisher Scientific | Cat# MA5-12960; RRID:AB_11000742 | 1:200 |
| Antibody | anti-Sarcomeric Alpha Actinin (Mouse Monoclonal) | Thermo Fisher Scientific | Cat# MA1-22863; RRID:AB_557426 | 1:200 |
| Antibody | anti-p63 (Mouse monoclonal) | Biocare Medical | Cat# CM163A; RRID:AB_10582730 | 1:100 |
| Antibody | anti-MUC5AC (Mouse monoclonal) | Thermo Fisher Scientific | Cat# MA5-12178; RRID:AB_10978001 | 1:100 |
| Antibody | anti-FOXJ1 (Mouse monoclonal) | Thermo Fisher Scientific | Cat# 14-9965-80; RRID:AB_1548836 | 1:100 |
| Antibody | anti-PAX8 (Mouse monoclonal) | ThermoFisher Scientific | Cat# MA1-117 RRID:AB_2536828 | 1:100 |
| Antibody | anti-β-Tubulin III (Mouse monoclonal) | Sigma-Aldrich | Cat# T8578 RRID:AB_1841228 | 1:100 |
| Antibody | anti-PAX6 (Mouse monoclonal) | BioLegends | Cat# 862,001 RRID:AB_2801237 | 1:100 |

*Appendix 1 Continued on next page*

*Appendix 1 Continued*

| Reagent type (species) or resource | Designation | Source or reference | Identifiers | Additional information |
|---|---|---|---|---|
| Antibody | anti-COUPTFII (Mouse monoclonal) | R&D Systems | Cat# PP-H7147-00 RRID:AB_1964214 | 1:100 |
| Antibody | anti-MLC2v (Rabbit Polyclonal) | ProteinTech Group | Cat# 10906–1-AP RRID:AB_2147453 | 1:100 |
| Antibody | anti-NFATC (Mouse monoclonal) | Thermo Fisher Scientific | Cat# MA3-024 RRID:AB_2236037 | 1:100 |
| Antibody | anti-WT1 (Mouse monoclonal) | Novus Biologicals | Cat# NBP2-44606 RRID:AB_not found | 1:100 |
| Antibody | anti-Brachyury (Goat polyclonal) | R&D Systems | Cat# AF2085; RRID:AB_2200235 | 1:50 |
| Antibody | anti-MIXL1 (Rabbit polyclonal) | Thermo Fisher Scientific | Cat# PA5-64903; RRID:AB_2664737 | 1:50 |
| Antibody | anti-NCAM1 (Rabbit monoclonal) | Cell Signaling Technologies | Cat# 99,746T; RRID:AB_2868490 | 1:50 |
| Antibody | anti-FOXA2 (Mouse monoclonal) | Santa Cruz Technology | Cat# Sc-271103; RRID:AB_10614496 | 1:50 |
| Antibody | anti-SOX17 (Goat polyclonal) | R&D Systems | Cat# AF1924; RRID:AB_355060 | 1:200 |
| Antibody | anti-OCT4 (Mouse monoclonal) | Santa Cruz | Cat# sc-5279 RRID:AB_628051 | 1:100 |
| Antibody | anti-CD13 APC-conjugated | BD Biosciences | Cat# 557454; RRID:AB_398624 | 1:10 |
| Antibody | Donkey anti-mouse IgG (H + L), Alexa Fluor 488 | Thermo Fisher Scientific | Cat# A21202; RRID:AB_141607 | 1:500 |
| Antibody | Donkey anti-rabbit IgG (H + L), Alexa Fluor 488 | Thermo Fisher Scientific | Cat# A21206; RRID:AB_2535792 | 1:500 |
| Antibody | Donkey anti-rabbit IgG (H + L), Alexa Fluor 568 | Thermo Fisher Scientific | Cat# A10042; RRID:AB_2757564 | 1:500 |
| Antibody | Donkey anti-goat IgG (H + L), Alexa Fluor 647 | Thermo Fisher Scientific | Cat# A21447; RRID:AB_141844 | 1:500 |
| Recombinant DNA protein | Activin A | R&D Systems | 338-AC-010 | |
| Recombinant DNA protein | Recombinant human BMP4 | R&D Systems | 314 BP | |
| Recombinant DNA protein | Recombinant human KGF | PeproTech | 100–19 | |
| Commercial assay, kit | High-Capacity cDNA Reverse Transcription kit | Applied Biosystems | 4368814 | |
| Commercial assay, kit | TaqMan Fast Advanced Master Mix | Thermo Fisher Scientific | 4444556 | |
| Commercial assay, kit | Fixable Violet Dead Cell Stain Kit | Thermo Fisher Scientific | L34955 | |
| Chemical compound, drugs | hESC-qualified Matrigel Basement Membrane Matrix | Corning | 354,234 | |
| Chemical compound, drugs | Growth Factor Reduced Basement Membrane Matrix | Corning | 354,230 | |
| Chemical compound, drugs | mTESR Plus | Stem Cell Technologies | 05825 | |
| Chemical compound, drugs | Dulbecco's Phosphate-Buffered Saline (DPBS) | Corning | 45000–430 | |

*Appendix 1 Continued on next page*

*Appendix 1 Continued*

| Reagent type (species) or resource | Designation | Source or reference | Identifiers | Additional information |
|---|---|---|---|---|
| Chemical compound, drugs | ReLESR | Stem Cell Technologies | 05873 | |
| Chemical compound, drugs | StemPro Accutase Cell Dissociation Reagent | Thermo Fisher Scientific | A1110501 | |
| Chemical compound, drugs | RPMI1640 | Corning | 10–040-CV | |
| Chemical compound, drugs | GlutaMAX | Thermo Fisher Scientific | 35050061 | |
| Chemical compound, drugs | B-27 minus insulin Supplement | Thermo Fisher Scientific | A1895601 | |
| Chemical compound, drugs | B-27 Supplement (Complete) | Thermo Fisher Scientific | 12587–010 | |
| Chemical compound, drugs | TrypLE Express | Thermo Fisher Scientific | 12605028 | |
| Chemical compound, drugs | Hyclone FetalClone 1 Serum (U.S) | GE Healthcare | SH30080.03 | |
| Chemical compound, drugs | Y-27632 dihydrochloride | Cayman Chemical | 1000558310 | |
| Chemical compound, drugs | CHIR99021 | Reprocell | 04000402 | |
| Chemical compound, drugs | A8301 | Sigma Aldrich | SSML1314-1MG | |
| Chemical compound, drugs | DMH-1 | Tocris | 4126/10 | |
| Chemical compound, drugs | IWP4 | Tocris | 5214/10 | |
| Chemical compound, drugs | All-trans Retinoic Acid | Cayman | 11,017 | |
| Chemical compound, drugs | Dexamethasone | Sigma Aldrich | D4902 | |
| Chemical compound, drugs | 8-bromoadenosine 3',5'-cyclic monophosphate sodium salt (cAMP) | Sigma Aldrich | B7880 | |
| Chemical compound, drugs | 3-Isobutyl-1-methylxanthine (IBMX) | Sigma Aldrich | I5879 | |
| Chemical compound, drugs | NSC668036 | Tocris | 5813/10 | |
| Chemical compound, drugs | PneumaCult-ALI Basal Medium | Stemcell Technologies | 05002 | |
| Chemical compound, drugs | PneumaCult-ALI Maintenance Supplement | Stemcell Technologies | 05006 | |
| Chemical compound, drugs | TRIzol Reagent | Thermo Fisher Scientific | 15596018 | |
| Chemical compound, drugs | Chloroform | Sigma-Aldrich | C2432 | |
| Chemical compound, drugs | Glycoblue | Thermo Fisher Scientific | AM9516 | |
| Chemical compound, drugs | Isopropanol | ACROS Organic | 327272500 | |
| Chemical compound, drugs | Ethanol 200 Proof | Pharmaco-AAPL | DSP-C7-18 | |

*Appendix 1 Continued on next page*

*Appendix 1 Continued*

| Reagent type (species) or resource | Designation | Source or reference | Identifiers | Additional information |
|---|---|---|---|---|
| Chemical compound, drugs | Methanol | Fisher Chemical | BPA412-1 | |
| Chemical compound, drugs | Paraformaldehyde | Sigma Aldrich | P6148-500G | |
| Chemical compound, drugs | Triton X-100 | Sigma Aldrich | X100-500ML | |
| Chemical compound, drugs | Bovine Serum Albumin | Fisher BioReagents | BP9706-100 | |
| Chemical compound, drugs | Phosphate Buffer Saline 20 X | Growcells | MRGF-695–010 L | |
| Chemical compound, drugs | Histoclear | Great Lakes | GL-1100–01 | |
| Chemical compound, drugs | Antigen Unmasking Solution, Citric Acid Based | Vector Laboratories | H-3300; RRID:AB_2336227 | |
| Chemical compound, drugs | DAPI-Fluoromount-G | Southern Biotech | 0100–20 | |
| Chemical compound, drugs | Hoechst 33,342 | Thermo Fisher Scientific | 62,249 | |
| Chemical compound, drugs | TrypLE Express Enzyme | Thermo Fisher Scientific | 12605010 | |
| Software | Image J | | Version 1.8.0.182; RRID:SCR_003070 | |
| Software | Flowjo | | Version 7.6.1; RRID:SCR_008520 | |
| Other | Transwell insert (0.4 µm) | Greiner Bio-One | 662,641 | |
| Other | Ultra-low adherence 24-well Plate | Greiner Bio-One | 662,970 | |
| Other | Ultra-low adherence 96-well Plate | Greiner Bio-One | 650,979 | |
| Other | Nanodrop 2000 Spectrophotometer | Thermo Fisher Scientific | ND2000CLAPTOP; RRID:SCR_018042 | |
| Other | EVOS FL Auto 2 Imaging System | Thermo Fisher Scientific | AMAFD2000 | |
| Other | CFX96 Touch Real-Time PCR Detection System | Bio-Rad | 1855196; RRID:SCR_018064 | |
| Other | ImmEdge Hydrophobic Barrier PAP Pen | Vector Laboratories | H-4000; RRID:AB_2336517 | |
| Other | HistoGel Specimen Processing Gel | Richard Allen Scientific | 11330057 | |

