## [Editor Report]

The study responded very well to the expert reviewers and offers new insights into mechanisms regulating differentiation of cardiac and pulmonary stem cells. It will stimulate further investigations into this important field.

---

## [Decision Letter]

**Decision letter after peer review:**

Thank you for submitting your article "Recapitulate Human Cardio-pulmonary Co-development Using Simultaneous Multilineage Differentiation of hPSCs" for consideration by *eLife*. Your article has been reviewed by 3 peer reviewers, and the evaluation has been overseen by a Reviewing Editor and Paul Noble as the Senior Editor. The following individual involved in review of your submission has agreed to reveal their identity: Sarah XL Huang (Reviewer #3).

The authors present a novel idea on simultaneous heart and lung induction from hPSCs. The work has a very good potential to be published in *eLife*. However, the manuscript in its current form is not publishable. There is consensus that the reproducibility of a given protocol must be demonstrated using multiple hPSC lines. Under COVID situation, the full protocol should be tested using at least another pluripotent cell line. In addition, the work in its current form lacks rigor and complete lineage characterization at key differentiation steps, as described in the review comments. The reviewers are confident that when such data are provided, there is a high possibility that it will support the conclusion of the paper, but the authors should not leave the readers to speculate.

The overall suggestion is that the authors should revise the manuscript to (1) perform additional thorough characterization of the differentiation culture up to the cardiac and lung progenitor specification stage of the protocol (day 15); (2) give an explicit clarification that progenitor specification is the focus of the current manuscript, and the maturation of cardiac and lung lineages will be investigated in-depth in near future studies. Otherwise, multi-lineage characterization within each of the organ and functional analyses on mature cell types are required for publication. This will take considerable time. (3) Analyses of the data that provide mechanistic insights into how each of the developmental milestones is achieved by the protocol (day 1-15), will greatly increase the impact of the paper.

*Reviewer #1 (Recommendations for the authors):*

1. Expression of lung (NKX2.1) and cardiac (NKX2.5) markers (Figure 1and2) is based on immunofluorescence and RT-PCR, but it remains unclear which percentage of cells expressed these markers, and which expressed neither marker. FACS analysis could have provided such data. This is relevant information for assessing the efficiency of the differentiation protocol (and comparison to those based on separate endoderm and mesoderm induction). Along the same line, it is unclear which percentage of cells ultimately expressed alveolar and cardiac markers in the 3D suspension cultures.

2. The approach to use scaffold-free 3D suspension cultures to create microtissues is not novel in the field of hiPSC and has been used e.g. in studies on cardiac myocyte and endothelial cell development. What was the reason to select this system, also because most alveolar differentiation protocols use Matrigel based organoid cultures?

3. In Figure 3, the authors use activin A to suppress development of cells of the cardiac lineage, and show that these support alveolar type 2 cell maturation. How did the authors check that activin A indeed suppressed development of cardiac cells and thus provide (indirect) evidence for the role of cardiac cells in alveolar development?

4. In Figure 3, the authors show that no alveolar type 2 cells are developed in the ALI culture system. Is there any evidence that this may be explained by loss of cells of the cardiac lineage?

5. Regarding further characterization of alveolar epithelial cells: did the authors also check for markers of type 1 alveolar epithelial cells?

6. Regarding further characterization of cardiac cells/cardiomyocytes: was any attempt made to quantify contraction, and show that this could e.g. be blocked using ca^2+^ channel blockers such as verapamil? And did they check other markers to show that the cells had developed into more mature heart cells, including other sarcomere markers, but also ion channels etc?

7. To achieve cardiopulmonary segregation as shown in Figure 4, the authors first used CKDCI medium from day 15-18, before switching to KDCI. What happened to alveolar type 2 marker expression upon CHIR withdrawal and segregation (was the alveolar niche fully established at day 18, and was therefore exogenous WNT activation no longer needed?), and was this segregation also observed if cultures were directly switched to KDCI on day 15?

*Reviewer #2 (Recommendations for the authors):*

While a potentially very interesting model, much more in depth characterization is required. Each component, as well intervening stages of development should be thoroughly characterized and benchmarked.

*Reviewer #3 (Recommendations for the authors):*

Here are further clarifications to some of the above-mentioned weaknesses, as well as additional suggestions on how the manuscript could be improved to increase the impact of the work.

Lines 24-30 of manuscript: the authors reasoned that WNT signaling is required for both mesodermal and endodermal specification. On this basis, the authors explained why they decided to use WNT agonist CHIR for mesoderm and DE co-induction from hiPSC line BU3. To my knowledge, the justification is not supported by lung developmental studies. In other words, the data shows convincing evidence of meso- and endo- dermal induction by the end of stage-1/day 4, however the interpretations provided by the authors are not accurate. A more plausible explanation is that, within the first 48 hours (day 1-2), high dose of WNT agonist CHIR induces primitive streak formation from part of the hiPSCs in culture. Subsequently, the newly formed primitive streak provides endogenous WNT, BMP and Nodal/Activin signaling to pattern the rest of the hiPSCs in culture to form mesoderm and DE by the end of day 4. This alternative mechanistic explanation is supported by evidence from both the existing literature and the authors' own data. Brivanlou's group elegantly demonstrated that activation of WNT signaling solely is sufficient to induce primitive streak from hPSCs. In addition, the dosage of CHIR applied in day 1-2 is way too high to be able to induce endoderm with lung potential. Furthermore, the authors maintain the cells in pluripotent medium mTESR1 for the first 48 hours (lines 48-50 of manuscript), which is critical for the success of the protocol (lines 87-90 of manuscript, suppl Figure 3). This supports my hypothesis that CHIR induces part of the hiPSCs in culture to form primitive streak during the first 48 hours, mTESR1 is required to maintain the remaining hiPS cells in culture in pluripotent state during this stage, before they start meso- and endo-dermal differentiation in day 2-4. Consistently, the data shows that Activin A or BMP inhibition during days 2-4 is detrimental for DE or mesodermal induction. Single cell analysis at 48 hr would be sufficient to prove a mixture of "primitive streak and pluripotent cells" profile, which will provide valuable mechanistic insights.

The whole manuscript is based on the differentiation of a double reporter line derived from one parental hiPSC line BU3. For scientific rigor and assurance, it is necessary to demonstrate efficient co-differentiation of lung and heart from other parental hPSC lines (both reporter and non-reporter) using the combination of time and dosage depicted in the current protocol.

It is convincing that most of the NKX2-1+ cells are lung fated and NKX2-5+ cells are cardiac fated, however, for scientific rigor and thoroughness, it is necessary to characterize the differentiation culture to exclude contamination of neural (EpCAM-) and thyroid (PAX8+) fated NKX2-1+ cells, and to confirm that all the NKX2-5+ cells co-express other cardiac progenitor markers such as KDR and PdgfR-α. Based on previous studies, some of the early lung progenitors co-express NKX2-5, therefore the authors must confirm whether NKX2-5+ cells at day 15 and on are NKX2-1- and EpCAM- cells.

[Editors’ note: further revisions were suggested prior to acceptance, as described below.]

Thank you for resubmitting your work entitled "Recapitulating Human Cardio-pulmonary Co-development Using Simultaneous Multilineage Differentiation of Pluripotent Stem Cells" for further consideration by *eLife*. Your revised article has been evaluated by Paul Noble (Senior Editor) and a Reviewing Editor.

The manuscript has been improved but there are some remaining issues that need to be addressed, as outlined below:

While the manuscript is improved, significant concerns remain that need to be addressed for publication. In particular, reviewer 2 has specific concerns that we would like to see addressed in a revision

*Reviewer #1 (Recommendations for the authors):*

The authors have made important changes and added results of new experiments in this extensively revised manuscript. This has markedly improved the quality of the manuscript, and conclusions are much better supported by the results.

As a result the revised manuscript is interesting and provides a new and relevant model to study heart and lung development in a dish, and cross-talk between these developing organs.

*Reviewer #2 (Recommendations for the authors):*

In this manuscript, Ng et al., report on a system where cardiac mesoderm and pulmonary endoderm co-develop from pluripotent stem cells. This is of potential interest, as it could provide an integrated model for the study of human cardiopulmonary development. I still have quite a few concerns, though, which were not fully addressed by the authors.

1. In my first review, I pointed out that the main weakness lies in the lack of thorough characterization of the resulting cells and tissues. This is still the case. The characterization relies almost entirely on reporter gene expression and PCR for the same markers. In Figure 1, representative examples are shown of a limited set of markers. It would be nice to have quantification (and for intracellular flow quantification, analysis of cells not expressing the protein of interest should be used as a control). The qPCR data do shown that these markers are expressed, but they compare something to nothing (i.e. PSCs), so that fold changes will be always be very high. One image of the ultrastructure of type 2 cells is shown now, which is a plus. Overall, this work still does begs the question why the only pulmonary markers observed in this model are basically SFTPC and NKX2.1. Furthermore, type 2 identity has not been further verified with markers such as SFTPB, ABCA3, LAMP2. Another outstanding question for the lung component is whether any pulmonary mesenchyme was generated. This was still not addressed. In this era, at scRNAseq should be performed. This would also address several of my other concerns below.

2. It is still not clear which cardiac cells are generated: ventricular, atrial, endocardium, epicardium, conducting tissue? This is the same comment as in my first review. An experiment with verapamil or some cells expressing α-actinin, as shown here, do not address this question. Atrial and ventricular cells contract. I would also add that in many PSC differentiation cultures where lineages are not sufficiently rigorously specified, contamination with beating cardiomyocytes is quite typical. This may be the case here. It is also difficult to understand why type 2 cells and unspecified cardiac cells are the only output of this model.

3. In Figure 2-S1d, the nuclear stains for NKX2.5 and NKX2.1 do not seem to align with DAPI. The same is true in Figure 3-S1a, where the higher magnifications for NKX2.1, NKX2.5 and DAPI do not seem to align. The same is true again for Figure 4-S1d, where the pattern of NKX2.1 is not the inverse of that of NK2.5 (no DAPI shown here). Furthermore, there is much more cTNT positivity than NKX2.5 expression in Figure 2-S1d. One would expect the opposite. These potential inconsistencies make the images difficult to interpret.

4. Along the same lines, in Figure 3a and 3d, as well in Figure 3-S1b and Figure 3-S3b (each time lower right panel), there seems to be co-expression of NKX2.1 and NKX2.5. Or is this superposition? Are there cells that express neither marker? This is difficult to evaluate without DAPI images. Also, could it be that in Figure 3-S1, the reduced cardiac and pulmonary differentiation is due to reduced proliferation? Again, without a DAPI image, this is difficult to evaluate.

5. This sentence is unclear: 'This may in part be due to the reduction of cardiac progenitor as indicated by significant downregulation of NKX2.5 gene expression (Figure 4f, Figure 4—figure supplement 1d).' (line 292). According to Figure 4f, there is no downregulation of NKX2.5 in suspension culture.

6. In Figure 4c, which relies entirely on reporters, it would be nice to co-stain for the actual proteins and other markers identifying type 2 cells. I note that in Figure 3c, the fluorescence patterns of the GFP and TdTomato reporters is very different, yet they should both be cytoplasmic. In the reporter studies, it would be good to co-differentiate a non-reporter expressing line as a reporter-negative control for autofluorescence in Figure 4 and its supplements. Figure 4-S1c is apparently not commented on the text.

7. The authors mention in the rebuttal evidence for type 1 cells, as shown by staining for HOPX in a non-reporter line (Figure 4-S5) and in the reporter line (Figure 6b). However, HOPX is not a marker for type 1 cells in humans, and is expressed broadly, including in cardiomyocytes (see, among others, Friedman et al., Cell Stem Cell 2018). In Figure 6b, there does not appear to be much pro-SFTPC, and that seems to be on the outside of the structure, whereas in Figures 4 and 5, the NKX2.1+ cells are on the inside.

8. The authors should not call Act. A 'TGF-β' signaling. It is a TGF-β family ligand, but signals through its own receptors. It recapitulates Nodal signaling in vitro.

9. Finally, there is still no sufficient quantification of differentiation efficiency and yield. NKX2.1, for example is also expressed in the forebrain and in the thyroid. The minimal conditions described typical yield neurectoderm. TUJ1 was examined, but other markers, such as PAX 6 should be included.

Recommendations for the authors

While a still potentially interesting model, much more in depth characterization is still required as well. In particular the nature and yield of the output should be better characterized, and figure quality and consistency should be paid attention to.

*Reviewer #3 (Recommendations for the authors):*

The authors addressed most of the questions and concerns raised in the 1st round of review.

---

## [Author Response]

The authors present a novel idea on simultaneous heart and lung induction from hPSCs. The work has a very good potential to be published in eLife. However, the manuscript in its current form is not publishable. There is consensus that the reproducibility of a given protocol must be demonstrated using multiple hPSC lines. Under COVID situation, the full protocol should be tested using at least another pluripotent cell line. In addition, the work in its current form lacks rigor and complete lineage characterization at key differentiation steps, as described in the review comments. The reviewers are confident that when such data are provided, there is a high possibility that it will support the conclusion of the paper, but the authors should not leave the readers to speculate.The overall suggestion is that the authors should revise the manuscript to (1) perform additional thorough characterization of the differentiation culture up to the cardiac and lung progenitor specification stage of the protocol (day 15); (2) give an explicit clarification that progenitor specification is the focus of the current manuscript, and the maturation of cardiac and lung lineages will be investigated in-depth in near future studies. Otherwise, multi-lineage characterization within each of the organ and functional analyses on mature cell types are required for publication. This will take considerable time. (3) Analyses of the data that provide mechanistic insights into how each of the developmental milestones is achieved by the protocol (day 1-15), will greatly increase the impact of the paper.

We appreciate the overall suggestions from the reviewers, which provide us valuable guidance to further improve this manuscript. Below we highlight key revisions in response to these overall suggestions.

(1) We have performed additional characterization of the differentiation up to Day-15. Specifically, we have included additional FACS analysis on Day-15 to provide a more comprehensive cellular characterization (Figure 2—figure supplement 1a), and have also confirmed that the NKX2.1^+^ progenitors are neither neural- nor thyroid-fated by co-staining with PAX8 and TUJ1 ( Figure 2—figure supplement 1d). Furthermore, we showed that NKX2.5^+^ progenitors are cardiac-fated as they co-express cTnT (Figure 2—figure supplement 1d).

(2) We agree with the reviewers’ comment and suggestion that the present study is primarily focused on progenitor specification. We have added explicit statements to emphasize this at discussion line 490-493; line 603-606. Nonetheless, in the revised manuscript, we have provided additional alveolar epithelial characterization in 3D cardio-pulmonary μTs to show positive immunofluorescence staining of SFTPC and HOPX (Figure 6b), as well as the presence of lamellar bodies by transmission electron microscopy (Figure 6a). In parallel, the cardiac μTs exhibited a striated pattern when stained for cTnT and sarcomeric α actinin (Figure 6c,d). Additionally, the cardiac μTs were functional and responded to calcium channel blockers (Verapamil) in a dose-dependent manner (Figure 6e).

(3) Based on the reviewers’ suggestions and to provide further mechanistic insights regarding developmental milestones, we have included staining markers of primitive streak (Brachyury) and mesendoderm (MIXL1) after 2 days of initial CHIR99021 induction, and used FACS analysis to show that the majority of resulting cells have committed to primitive streak at this stage (Figure 1—figure supplement 1).

(4) We have also validated our differentiation protocol on an additional hiPSC line, including germ layer induction, cardio-pulmonary progenitor induction, and 3D µT formation and maturation.

Reviewer #1 (Recommendations for the authors):1. Expression of lung (NKX2.1) and cardiac (NKX2.5) markers (Figure 1and2) is based on immunofluorescence and RT-PCR, but it remains unclear which percentage of cells expressed these markers, and which expressed neither marker. FACS analysis could have provided such data. This is relevant information for assessing the efficiency of the differentiation protocol (and comparison to those based on separate endoderm and mesoderm induction). Along the same line, it is unclear which percentage of cells ultimately expressed alveolar and cardiac markers in the 3D suspension cultures.

Thank you for bringing this to our attention. We have included FACS analysis of pulmonary progenitors (based on their NKX2.1-driven GFP expression) and cardiac progenitors (NKX2.5 antibody labeling) (Figure 1—figure supplement 1a). In addition, we have included the FACS analysis of endoderm-focused pulmonary protocol and mesoderm-focused cardiac protocol (Figure 1—figure supplement 1b, c) for comparison. We have also provided further characterization of alveolar and cardiac marker expression in 3D suspension culture (Figure 6).

2. The approach to use scaffold-free 3D suspension cultures to create microtissues is not novel in the field of hiPSC and has been used e.g. in studies on cardiac myocyte and endothelial cell development. What was the reason to select this system, also because most alveolar differentiation protocols use Matrigel based organoid cultures?

As the reviewer has pointed out, most current protocols for inducing alveolar differentiation utilize Matrigel embedding. ^1,2^ To address the reviewer’s comment, we have added a new figure (Figure 4—figure supplement 3) comparing alveolar type 2 cell induction in our 3D suspension culture versus Matrigel-embedded culture from Day-15 cardio-pulmonary progenitors, demonstrating that 3D suspension culture on ultra-low adhesion surface expedites alveolar type 2 maturation on Day-18 of differentiation (Day-3 of alveolar maturation).

3. In Figure 3, the authors use activin A to suppress development of cells of the cardiac lineage, and show that these support alveolar type 2 cell maturation. How did the authors check that activin A indeed suppressed development of cardiac cells and thus provide (indirect) evidence for the role of cardiac cells in alveolar development?

In Figure 3a-c, we showed that the presence of Activin A in our cardio-pulmonary protocol significantly downregulated NKX2.5 gene expression (by qPCR), and this was further supported by immunostaining (Figure 3a). To further confirm that there was no cardiac population in Activin A-supplemented protocol, we generated 3D μTs from Day-15 progenitors and showed that no NKX2.5 expression could be observed using confocal microscopy (Figure 4l). Therefore, we confirmed that Activin A supplementation suppressed the induction of cardiac cells, which suggested the potential positive impact of cardiac accompaniment during alveolar maturation. This is consistent with findings from rodent embryogenesis that chemical cues derived from the heart field play key roles in lung development. ^3,4^ We have included additional discussion in the revised manuscript (line 556-559).

4. In Figure 3, the authors show that no alveolar type 2 cells are developed in the ALI culture system. Is there any evidence that this may be explained by loss of cells of the cardiac lineage?

We would like to clarify that alveolar type 2 cells were still being developed in the ALI culture but to a very minimal extent compared to 3D suspension culture (Figure 4c). In the revised manuscript, we have added qPCR analysis to compare cardiac NKX2.5 expression in both ALI and 3D suspension culture. We found that NKX2.5 gene expression (by qPCR) was significantly downregulated in ALI culture compared to 3D suspension culture (Figure 4f). This was further supported by immunostaining (Figure 4—figure supplement 1d), which showed a reduction in the cardiac population in ALI culture.

5. Regarding further characterization of alveolar epithelial cells: did the authors also check for markers of type 1 alveolar epithelial cells?

Thank you for pointing this out. Based on the reviewer’s suggestion, in the revised manuscript, we added alveolar type 1 (AT1) cell characterization by performing HOPX staining ^5-7^ on Day-22 cardio-pulmonary μTs and observed the presence of AT1-like cells that did not colocalize with the AT2 cell marker SFTPC (Figure 6b). Nonetheless, the organization of these AT1-like cells in μTs remained different from that in the native lung. This suggests that further optimization of AT1 maturation is needed to overcome this limitation. We have also addressed this in the manuscript (line 585-587).

6. Regarding further characterization of cardiac cells/cardiomyocytes: was any attempt made to quantify contraction, and show that this could e.g. be blocked using ca^2+^ channel blockers such as verapamil? And did they check other markers to show that the cells had developed into more mature heart cells, including other sarcomere markers, but also ion channels etc?

To further investigate the function of contracting cardiac lineages, in the revised manuscript, we added an experiment to treat the cardio-pulmonary μTs with different concentrations of Verapamil (calcium channel blocker). Using Ca520 as a calcium influx indicator, we showed that the effect of Verapamil on the cardiac μTs was dose-dependent (Figure 6e). Furthermore, we have included cTnT and Sarcomeric Α Actinin staining on Day-22 μTs, which revealed a striated pattern (Figure 6c,d). Together, this provides more compelling evidence of the functionality of the induced cardiac tissues.

7. To achieve cardiopulmonary segregation as shown in Figure 4, the authors first used CKDCI medium from day 15-18, before switching to KDCI. What happened to alveolar type 2 marker expression upon CHIR withdrawal and segregation (was the alveolar niche fully established at day 18, and was therefore exogenous WNT activation no longer needed?), and was this segregation also observed if cultures were directly switched to KDCI on day 15?

In the revised manuscript, we have included a new figure showing that removal of CHIR on Day-18 led to a decrease in SFTPC expression (Figure 4—figure supplement 1c), indicating that the alveolar niche was not fully established by Day-18. Further investigation is needed to address whether the complete supporting niche for alveologenesis can be fully established and if the corresponding time point can be identified. In addition, we found that CHIR was indeed an important factor to induce alveologenesis, consistent with previous publications. ^1,2^ Without the addition of CHIR following Day-15 differentiation, we observed no maturation of AT2 cells on Day-18 of µT culture (Figure 5—figure supplement 1a,b,c). As for segregation, cardio-pulmonary µTs cultured directly in KDCI (without CHIR) from Day-15 were still able to undergo segregation over time (Figure 5—figure supplement 1d,e). We have included additional discussion regarding the findings and limitations of the present study (line 595-601).

Reviewer #2 (Recommendations for the authors):While a potentially very interesting model, much more in depth characterization is required. Each component, as well intervening stages of development should be thoroughly characterized and benchmarked.

Thank you for the comprehensive comments on our manuscript, we have performed an extensive revision as described above in response to the reviewer’s comments.

Reviewer #3 (Recommendations for the authors):Here are further clarifications to some of the above-mentioned weaknesses, as well as additional suggestions on how the manuscript could be improved to increase the impact of the work.Lines 24-30 of manuscript: the authors reasoned that WNT signaling is required for both mesodermal and endodermal specification. On this basis, the authors explained why they decided to use WNT agonist CHIR for mesoderm and DE co-induction from hiPSC line BU3. To my knowledge, the justification is not supported by lung developmental studies. In other words, the data shows convincing evidence of meso- and endo- dermal induction by the end of stage-1/day 4, however the interpretations provided by the authors are not accurate. A more plausible explanation is that, within the first 48 hours (day 1-2), high dose of WNT agonist CHIR induces primitive streak formation from part of the hiPSCs in culture. Subsequently, the newly formed primitive streak provides endogenous WNT, BMP and Nodal/Activin signaling to pattern the rest of the hiPSCs in culture to form mesoderm and DE by the end of day 4. This alternative mechanistic explanation is supported by evidence from both the existing literature and the authors' own data. Brivanlou's group elegantly demonstrated that activation of WNT signaling solely is sufficient to induce primitive streak from hPSCs. In addition, the dosage of CHIR applied in day 1-2 is way too high to be able to induce endoderm with lung potential. Furthermore, the authors maintain the cells in pluripotent medium mTESR1 for the first 48 hours (lines 48-50 of manuscript), which is critical for the success of the protocol (lines 87-90 of manuscript, suppl Figure 3). This supports my hypothesis that CHIR induces part of the hiPSCs in culture to form primitive streak during the first 48 hours, mTESR1 is required to maintain the remaining hiPS cells in culture in pluripotent state during this stage, before they start meso- and endo-dermal differentiation in day 2-4. Consistently, the data shows that Activin A or BMP inhibition during days 2-4 is detrimental for DE or mesodermal induction. Single cell analysis at 48 hr would be sufficient to prove a mixture of "primitive streak and pluripotent cells" profile, which will provide valuable mechanistic insights.

We thank the reviewer for sharing the comprehensive opinion. We agree that endogenous signaling is the key to promote differentiation of both mesoderm and endoderm in our culture. In the revised manuscript, we have included new FACS data for T (Brachyury) and MIXL1 after 2 days of CHIR treatment and found that the majority of cells were T and MIXL1 positive (primitive streak/mesendoderm) (Figure 1—figure supplement 1d,e). Furthermore, we performed analysis of pluripotent cell marker (OCT4) and did not observe any remaining pluripotent cells after exposure to CHIR for 2 days (Figure 1—figure supplement 1c). Lastly, we appreciate the reviewer for bringing the related literature to our attention and have included the references as well as the related discussion to the revised manuscript (line 522-524).

The whole manuscript is based on the differentiation of a double reporter line derived from one parental hiPSC line BU3. For scientific rigor and assurance, it is necessary to demonstrate efficient co-differentiation of lung and heart from other parental hPSC lines (both reporter and non-reporter) using the combination of time and dosage depicted in the current protocol.

Thank you for pointing this out. In the revised manuscript, we have further showed that the presented protocol can be reproduced in another independent hiPSC line (BU1), as indicated by differentiation into cardiopulmonary progenitors on Day-15. Furthermore, we have included additional data for BU1 cell line regarding mesoderm and endoderm induction during Stage-1 (Figure 4—figure supplement 5b) and cardio-pulmonary μT formation from Day-15 progenitor cells. On Day-18, the BU1-derived dual-lineage μTs were positive for NKX2.1 and NKX2.5, similar to what we observed in BU3. Furthermore, the μTs were stained positive for SFTPC and HOPX, indicative of their potential for further alveolar maturation. Additionally, the BU1-derived NKX2.5 cardiac lineages co-expressed cTnT and Sarcomeric Α Actinin (Figure 4—figure supplement 5e).

It is convincing that most of the NKX2-1+ cells are lung fated and NKX2-5+ cells are cardiac fated, however, for scientific rigor and thoroughness, it is necessary to characterize the differentiation culture to exclude contamination of neural (EpCAM-) and thyroid (PAX8+) fated NKX2-1+ cells, and to confirm that all the NKX2-5+ cells co-express other cardiac progenitor markers such as KDR and PdgfR-α. Based on previous studies, some of the early lung progenitors co-express NKX2-5, therefore the authors must confirm whether NKX2-5+ cells at day 15 and on are NKX2-1- and EpCAM- cells.

To exclude the possibility of neural- and thyroid-fate of NKX2.1^+^ cells, in the revised manuscript, we have included the staining of TUJ1 (neural) and PAX8 (thyroid) on Day-15 differentiated cells. We did not observe any co-localization between either TUJ1 or PAX8 with NKX2.1 (Figure 2—figure supplement 2d). To confirm the identity of NKX2.5^+^ cells, we performed immunostaining and demonstrated their colocalization with cTnT, further confirming their cardiac fate. Regarding possible co-expression of NKX2.1 and NKX2.5, please refer to whole mount images in Figure 4b and Figure 4i. We have carefully checked each confocal section and did not observe colocalization between the two markers.

References:

1. Jacob, A. et al. Differentiation of Human Pluripotent Stem Cells into Functional Lung Alveolar Epithelial Cells. Cell stem cell 21, 472-488, doi:10.1016/j.stem.2017.08.014 (2017).

2. Huang, S. X. L. et al. Efficient generation of lung and airway epithelial cells from human pluripotent stem cells. Nat Biotech 32, 84-91, doi:10.1038/nbt.2754 (2014).

3. Steimle, J. D. et al. Evolutionarily conserved Tbx5-Wnt2/2b pathway orchestrates cardiopulmonary development. Proceedings of the National Academy of Sciences 115, E10615-E10624, doi:10.1073/pnas.1811624115 (2018).

4. Peng, T. et al. Coordination of heart and lung co-development by a multipotent cardiopulmonary progenitor. Nature 500, 589-592, doi:10.1038/nature12358 (2013).

5. Wang, Y. et al. Pulmonary alveolar type I cell population consists of two distinct subtypes that differ in cell fate. Proceedings of the National Academy of Sciences of the United States of America 115, 2407-2412, doi:10.1073/pnas.1719474115 (2018).

6. Frank, D. B. et al. Early lineage specification defines alveolar epithelial ontogeny in the murine lung. Proceedings of the National Academy of Sciences 116, 4362, doi:10.1073/pnas.1813952116 (2019).

7. Liebler, J. M. et al. Combinations of differentiation markers distinguish subpopulations of alveolar epithelial cells in adult lung. American journal of physiology. Lung cellular and molecular physiology 310, L114-120, doi:10.1152/ajplung.00337.2015 (2016).

8. Wang, S. et al. GSK-3beta Inhibitor CHIR-99021 Promotes Proliferation Through Upregulating β-Catenin in Neonatal Atrial Human Cardiomyocytes. Journal of cardiovascular pharmacology 68, 425-432, doi:10.1097/fjc.0000000000000429 (2016).

9. Tseng, A. S., Engel, F. B. and Keating, M. T. The GSK-3 inhibitor BIO promotes proliferation in mammalian cardiomyocytes. Chemistry and biology 13, 957-963, doi:10.1016/j.chembiol.2006.08.004 (2006).

10. Buikema, J. W. et al. Wnt Activation and Reduced Cell-Cell Contact Synergistically Induce Massive Expansion of Functional Human iPSC-Derived Cardiomyocytes. Cell stem cell 27, 50-63.e55, doi:10.1016/j.stem.2020.06.001 (2020).

[Editors' note: further revisions were suggested prior to acceptance, as described below.]

While the manuscript is improved, significant concerns remain that need to be addressed for publication. In particular, reviewer 2 has specific concerns that we would like to see addressed in a revisionReviewer #1 (Recommendations for the authors):The authors have made important changes and added results of new experiments in this extensively revised manuscript. This has markedly improved the quality of the manuscript, and conclusions are much better supported by the results.As a result the revised manuscript is interesting and provides a new and relevant model to study heart and lung development in a dish, and cross-talk between these developing organs.

Thank you for the comment. In the revised manuscript (line 473-475), we have provided additional discussion regarding HOPX expression and highlighted the need of future investigations of alveolar maturation (in particular towards AT1 fate) in a more comprehensive manner. We have followed the overall suggestion from the reviewers and editor and have provided an explicit clarification that progenitor specification is the focus of the current manuscript.

Reviewer #2 (Recommendations for the authors):In this manuscript, Ng et al., report on a system where cardiac mesoderm and pulmonary endoderm co-develop from pluripotent stem cells. This is of potential interest, as it could provide an integrated model for the study of human cardiopulmonary development. I still have quite a few concerns, though, which were not fully addressed by the authors.1. In my first review, I pointed out that the main weakness lies in the lack of thorough characterization of the resulting cells and tissues. This is still the case. The characterization relies almost entirely on reporter gene expression and PCR for the same markers. In Figure 1, representative examples are shown of a limited set of markers. It would be nice to have quantification (and for intracellular flow quantification, analysis of cells not expressing the protein of interest should be used as a control). The qPCR data do shown that these markers are expressed, but they compare something to nothing (i.e. PSCs), so that fold changes will be always be very high. One image of the ultrastructure of type 2 cells is shown now, which is a plus. Overall, this work still does begs the question why the only pulmonary markers observed in this model are basically SFTPC and NKX2.1. Furthermore, type 2 identity has not been further verified with markers such as SFTPB, ABCA3, LAMP2. Another outstanding question for the lung component is whether any pulmonary mesenchyme was generated. This was still not addressed. In this era, at scRNAseq should be performed. This would also address several of my other concerns below.

We thank the reviewer for the suggestions and have organized our response in three subsections as described below.

(a) It would be nice to have quantification (and for intracellular flow quantification), analysis of cells not expressing the protein of interest should be used as a control.

We have performed additional flow cytometry experiments to characterize and quantitate mesoderm and endoderm induction during Stage-1 of differentiation. Our data showed 47.0% of CD13-expressing mesodermal cells and 45.2% of SOX17-expressing endodermal cells (Figure 1—figure supplement 1f-g). We have also included hiPSCs as our control, which expressed neither of these markers.

(b) The qPCR data do shown that these markers are expressed, but they compare something to nothing (i.e. PSCs), so that fold changes will be always be very high.

We thank the reviewer to point this out. In figures such as Figure 1d, we agree that normalization based on hiPSCs that did not express the target gene led to fold changes that appeared to be very high. However, the main comparisons that we intend to deliver in these experiments were those between different differentiation conditions, such as different CHIR concentrations, where in all conditions there were measurable expression of the target gene and the fold change between groups is much more realistic than the comparison versus hiPSCs. The high fold change values relative to hiPSC control did not affect us to perform accurate comparison regarding gene expression levels between different differentiation conditions.

(c) Overall, this work still does begs the question why the only pulmonary markers observed in this model are basically SFTPC and NKX2.1. furthermore, type 2 identity has not been further verified with markers such as SFTPB, ABCA3, LAMP2. Another outstanding question for the lung component is whether any pulmonary mesenchyme was generated.

In this manuscript, we used CKDCI (CHIR99021, KGF, dexamethasone, cAMP, and IBMX) maturation medium that have been shown to selectively promote differentiation of alveolar type 2 (AT2) cells of the distal lung [1, 2]. Further, based on the reviewer’s suggestion, we have included characterization of an additional marker (pro-SFTPB) to verify AT2 identity (Figure 6—figure supplement 1a). Staining of S100A4, a marker for mesenchyme, has been performed in the 3D cardio-pulmonary μT (Figure 6—figure supplement 1c).

2. It is still not clear which cardiac cells are generated: ventricular, atrial, endocardium, epicardium, conducting tissue? This is the same comment as in my first review. An experiment with verapamil or some cells expressing α-actinin, as shown here, do not address this question. Atrial and ventricular cells contract. I would also add that in many PSC differentiation cultures where lineages are not sufficiently rigorously specified, contamination with beating cardiomyocytes is quite typical. This may be the case here. It is also difficult to understand why type 2 cells and unspecified cardiac cells are the only output of this model.

(a) It is still not clear which cardiac cells are generated.

In the revised manuscript, we have performed additional staining to characterize the identity of the induced cardiac cells, including ventricular myocytes (MLC2v), atrial myocytes (COUPTFII), endocardium (NFATC) and epicardium (WT1). In Figure 2—figure supplement 1d, we observed presence of NKX2.5+COUPTFII+ cardiac cells but negative for the rest markers such as MLC2v, NFATC and WT1, suggesting that the some of the cardiac cells have been specified into atrial myocytes. The observation of atrial induction is consistent with the presence of retinoic acid in our co-differentiation medium, which has been shown to be critical for atrial specification [3-5]. Further induction of other cardiac cell lineages is expected to require specifically formulated maturation medium and extended maturation time.

(b) Atrial and ventricular cells contract. I would also add that in many PSC differentiation cultures where lineages are not sufficiently rigorously specified, contamination with beating cardiomyocytes is quite typical. This may be the case here.

We agree to the reviewer’s point that minor cardiac population could emerge during differentiation where mesodermal cells are permitted. However, based on our flow cytometry analysis (Figure 2—figure supplement 1a), a balanced induction of pulmonary (32.8%) and cardiac (39.3%) cells were observed. In Figure 2—figure supplement 1b, we further confirmed that emergence of cardiac cells could be modulated by Activin A. Thus, the emergence of cardiac lineage in the presented co-differentiation system was a controlled process.

(c) It is also difficult to understand why type 2 cells and unspecified cardiac cells are the only output of this model.

In this manuscript, we used CKDCI (CHIR99021, KGF, dexamethasone, cAMP, and IBMX) maturation medium that have been shown to selectively promote differentiation of alveolar type 2 (AT2) cells of the distal lung [1, 2, 6]. The same goes for cardiac subtype specification. The observation of atrial induction is consistent with the presence of retinoic acid in our co-differentiation medium, which has been shown to be critical for atrial specification [3-5]. Further induction of other cardiac cell lineages is expected to require specifically formulated maturation medium and extended maturation time.

3. In Figure 2-S1d, the nuclear stains for NKX2.5 and NKX2.1 do not seem to align with DAPI. The same is true in Figure 3-S1a, where the higher magnifications for NKX2.1, NKX2.5 and DAPI do not seem to align. The same is true again for Figure 4-S1d, where the pattern of NKX2.1 is not the inverse of that of NK2.5 (no DAPI shown here). Furthermore, there is much more cTNT positivity than NKX2.5 expression in Figure 2-S1d. One would expect the opposite. These potential inconsistencies make the images difficult to interpret.

Due to high confluency after long term culture (15 days), the cells were arranged in multiple layers, resulting in superimposition, and thus it was difficult to appreciate DAPI staining with the presence of different cell lineages. Although confocal imaging would have solved this issue, but the co-differentiation was performed in 96 well plate, rendering difficulty to image with confocal microscopy. We have included respective DAPI images in Figure 2—figure supplement 1d, Figure 3—figure supplement 1a, and Figure 4—figure supplement 1d. Figure 4-S1d represented the cells cultured on Transwell Membrane. While one may expect the inverse relationship between NKX2.1 and NKX2.5 staining, within Day-15 differentiated cells, there were still about 20% of cell that belong to neither cardiac or pulmonary lineages. In Figure 2—figure supplement 1d, we observed that majority of the NKX2.5 cells were cTnT positive.

4. Along the same lines, in Figure 3a and 3d, as well in Figure 3-S1b and Figure 3-S3b (each time lower right panel), there seems to be co-expression of NKX2.1 and NKX2.5. Or is this superposition? Are there cells that express neither marker? This is difficult to evaluate without DAPI images. Also, could it be that in Figure 3-S1, the reduced cardiac and pulmonary differentiation is due to reduced proliferation? Again, without a DAPI image, this is difficult to evaluate.

We thank the reviewer for the comment. In Figure 3a, 3d, Figure 3—figure supplement 1b, Figure 3—figure supplement 3b, due to high cell confluency, the resulting cells from different lineages were superimposed. As the reviewer suggested, we have included DAPI images. By referring to Figure 4b, using confocal imaging of cardio-pulmonary μT, we did not observe obvious co-localization between NKX2.1 and NKX2.5 expression. Indeed, there were cells that do not express either cardiac or pulmonary marker as shown by flow cytometry analysis, where there were about 20% cells that expressed neither NKX2.1 nor NKX2.5 (Figure 2—figure supplement 1a). In Figure 3—figure supplement 1, it is still possible that the reduced cardio-pulmonary differentiation was due to reduced proliferation since initial differentiation medium in B27 minus insulin was harsh compared to mTESR1. However, the well was full of cells at the end of differentiation.

5. This sentence is unclear: 'This may in part be due to the reduction of cardiac progenitor as indicated by significant downregulation of NKX2.5 gene expression (Figure 4f, Figure 4—figure supplement 1d).' (line 292). According to Figure 4f, there is no downregulation of NKX2.5 in suspension culture.

Thank you for the comment. The sentence “This may in part be due to the reduction of cardiac progenitor as indicated by significant downregulation of NKX2.5 gene expression (Figure 4f, Figure 4—figure supplement 1d).” In this context, we were comparing 3D suspension culture with ALI culture. We found 3D suspension is a more robust system for alveolar maturation as indicated by SFTPC gene induction. We anticipated that the lower alveolar specification efficiency in ALI culture was due to reduction of cardiac progenitor as pointed out in Figure 4f (ALI vs. Day-15 cells) and Figure 4—figure supplement 1d (Immunostaining on ALI) (line 295).

6. In Figure 4c, which relies entirely on reporters, it would be nice to co-stain for the actual proteins and other markers identifying type 2 cells. I note that in Figure 3c, the fluorescence patterns of the GFP and TdTomato reporters is very different, yet they should both be cytoplasmic. In the reporter studies, it would be good to co-differentiate a non-reporter expressing line as a reporter-negative control for autofluorescence in Figure 4 and its supplements. Figure 4-S1c is apparently not commented on the text.

We thank the reviewer for pointing this out. We have included staining for both pro-SFTPC and pro-SFTPB to further verify type 2 identify as demonstrated in Figure 6b and Figure 6—figure supplement 1a. The fluorescence reporter TdTomato is generally a subset of GFP, as only a portion of the GFP+ lung progenitors have matured into type 2 cells (labelled by TdTomato). Thus, not all GFP cells would express SFTPC. To address the possibility of autofluorescence, as the reviewer suggested, we have added representative images of non-reporter BU1 hiPSC line as a control (Figure 4—figure supplement 6). We have also commented Figure 4—figure supplement 1c in the text (line 286-287).

7. The authors mention in the rebuttal evidence for type 1 cells, as shown by staining for HOPX in a non-reporter line (Figure 4-S5) and in the reporter line (Figure 6b). However, HOPX is not a marker for type 1 cells in humans, and is expressed broadly, including in cardiomyocytes (see, among others, Friedman et al., Cell Stem Cell 2018). In Figure 6b, there does not appear to be much pro-SFTPC, and that seems to be on the outside of the structure, whereas in Figures 4 and 5, the NKX2.1+ cells are on the inside.

(a) HOPX is not a marker for type 1 cells in humans, and is expressed broadly, including in cardiomyocytes.

Thank you for the comment. We agree that HOPX alone does not fully define AT1 cells. In the revised manuscript (line 473-475), we have provided additional discussion regarding HOPX expression and highlighted the need of future investigations of alveolar maturation (in particular towards AT1 fate) in a more comprehensive manner. Although HOPX expression has been observed in cardiomyocytes, [7] we did not observe HOPX expression following establishment of cardio-pulmonary progenitors, and the induction of HOPX coincided with alveolar maturation during Day-15 to Day-18.

(b) In Figure 6b, there does not appear to be much pro-SFTPC, and that seems to be on the outside of the structure, whereas in Figures 4 and 5, the NKX2.1+ cells are on the inside.

In Figure 6b, the μTs have been cultured in maturation medium without CHIR for 4 days. With CHIR withdrawal after Day-18, tissues in the μTs would have reorganized themselves and the pulmonary tissue started to undergo segregation from the cardiac tissue, leading to the pulmonary lineages to move towards the edge (Figure 5a). While in Figure 4, and Figure 5, the μTs were of Day-18 (before cardio-pulmonary segregation).

8. The authors should not call Act. A 'TGF-β' signaling. It is a TGF-β family ligand, but signals through its own receptors. It recapitulates Nodal signaling in vitro.

Thanks for the recommendation. We have changed the term “TGF-β signaling” to “nodal signaling” for activin A.

9. Finally, there is still no sufficient quantification of differentiation efficiency and yield. NKX2.1, for example is also expressed in the forebrain and in the thyroid. The minimal conditions described typical yield neurectoderm. TUJ1 was examined, but other markers, such as PAX 6 should be included.

The yield of NKX2.1- and NKX2.5-expressing cells on Day-15 cells were as shown in Figure 2—figure supplement 1a. We have also demonstrated that NKX2.1+ cells on Day-15 do not co-express another forebrain marker such as PAX6. The data has been included in Figure 2—figure supplement 1d.

Recommendations for the authorsWhile a still potentially interesting model, much more in depth characterization is still required as well. In particular the nature and yield of the output should be better characterized, and figure quality and consistency should be paid attention to.

Thank you for the comprehensive suggestions, and we have revised the manuscript to address comments as described above.

References:

1. Huang, S.X.L., et al., Efficient generation of lung and airway epithelial cells from human pluripotent stem cells. Nat Biotech, 2014. 32(1): p. 84-91.

2. Jacob, A., et al., Differentiation of Human Pluripotent Stem Cells into Functional Lung Alveolar Epithelial Cells. Cell Stem Cell, 2017. 21(4): p. 472-488.e10.

3. Zhang, Q., et al., Direct differentiation of atrial and ventricular myocytes from human embryonic stem cells by alternating retinoid signals. Cell Research, 2011. 21(4): p. 579-587.

4. Miao, S., et al., Retinoic acid promotes metabolic maturation of human Embryonic Stem Cell-derived Cardiomyocytes. Theranostics, 2020. 10(21): p. 9686-9701.

5. Gassanov, N., et al., Retinoid acid-induced effects on atrial and pacemaker cell differentiation and expression of cardiac ion channels. Differentiation, 2008. 76(9): p. 971-80.

6. Hawkins, F.J., et al., Derivation of Airway Basal Stem Cells from Human Pluripotent Stem Cells. Cell Stem Cell, 2021. 28(1): p. 79-95.e8.

7. Friedman, C.E., et al., Single-Cell Transcriptomic Analysis of Cardiac Differentiation from Human PSCs Reveals HOPX-Dependent Cardiomyocyte Maturation. Cell stem cell, 2018. 23(4): p. 586-598.e8.

8. Hogan, B.L., Bone morphogenetic proteins in development. Curr Opin Genet Dev, 1996. 6(4): p. 432-8.

9. Schier, A.F., Nodal signaling in vertebrate development. Annu Rev Cell Dev Biol, 2003. 19: p. 589-621.

10. Yamaguchi, T.P., Heads or tails: Wnts and anterior-posterior patterning. Curr Biol, 2001. 11(17): p. R713-24.